# Disequilibrium response to tapping crustal magma reveals storage conditions

Janine Birnbaum[1 ✉], Fabian B. Wadsworth[1], Jackie E. Kendrick[1], Ben Kennedy[2], Paul A. Wallace[1], Marize Muniz da Silva[1], Kai-Uwe Hess[1] & Yan Lavallée[1]

The conditions under which magma accumulates and is stored are fundamental to unravelling the processes of crust formation, planetary differentiation, geothermal heat recharge and volcanic eruptions. Storage pressure, temperature and volatile saturation are typically inferred from erupted volcanic products. However, changes during kilometres of magma ascent induce disequilibrium crystallization and vesiculation, and inverting back to storage conditions comes with unresolvable uncertainties. Here we explore opportunities arising from magma drilling at Krafla volcano, Iceland, to reconstruct real, in situ magmatic conditions. The findings show that, over the approximately 5 min in which the magma is quenched, vapour bubbles consisting of $H_2O$ and $CO_2$ exsolve, grow and resorb, but the changes can be accounted for by multiparametric inversion (for chemistry, vesicularity and vitrification), and that the magma was stored under volatile-saturated lithostatic conditions, unlike previous assertions of lower vapour pressures based on classic methods[1]. These new disequilibrium simulations reconcile the glass chemistry with conceptual models of magma storage and provide us with the unique pairing of precisely measured depth and volatile pressure on a single magma body and thus a robust method to improve our understanding of magma storage conditions and evolution.

Despite their importance, estimates of the pressure, temperature, saturation state, geometry and location of magmatic storage regions vary widely for even the most-studied individual volcanic systems. Geothermo-barometers using mineral–mineral chemistry or phase equilibria have been applied to volcanic materials to constrain magmatic origins, but these methods have large uncertainties (about 50–200 MPa or about 2.5–10.0 km) arising from sparse laboratory constraints, limits on analytical precision, assumptions of local equilibrium and interdependence between temperature and pressure[2,3]. Furthermore, magmatic reservoirs are challenging to image with geophysical methods owing to limitations in resolution and poorly constrained relationships between lithology and geophysical signals, resulting in typical uncertainties on magma depth of approximately 0.5–10.0 km (ref. 4). As a result, the scientific community lacks consensus on even the fundamentals of the spatial distribution of melt in the crust[5].

Deep drilling in hydrothermal fields offers the unique potential for placing tight constraints on the location, temperature, pressure and chemistry of melt stored in natural volcanic systems. Hydrothermal drilling has occasionally intersected magma: dacite in the Puna geothermal field (Hawaii), trachyte at Menengai (Kenya) and rhyolite at Krafla (Iceland). At Krafla, the KJ-39 and Iceland Deep Drilling Project-1 (IDDP-1) boreholes directly intersected magma at roughly 2,500 and at 2,104 m depth, respectively[6,7], which was not anticipated on the basis of coarse geophysical imaging before drilling. Retrospectively, a magma body was recognized during reanalysis of magnetotelluric data[8] and from

seismic imaging[9,10], which have still been unable to resolve questions about its lateral and vertical extent.

Silicic glass fragments were recovered from the IDDP-1 borehole, which were quenched through interaction with drilling fluids[7,11]. The precisely known recovery depth, and temporally and spatially constrained ascent and quench, makes them ideally suited to resolve the unknowns of magmatic storage, as the melt has not been subject to the complex ascent processes that afflict the products of volcanic eruptions. Indeed, the glass chemistry has sparked discussion about the origins of the magma from (1) partially molten, hydrothermally altered basaltic crust[12] or (2) mantle-derived basalts evolved by means of fractional crystallization[13] and about the degree of crustal assimilation[1,12,14,15]. Despite the direct sampling of the glass, established equilibrium-based methods to determine pressure and temperature have yielded wide constraints for these magmas (Fig. 1a): various geothermometers yield different estimates and uncertainties from two-pyroxene equilibration between 920 °C and 940 °C (refs. 1,16) and 890 °C and 910 °C (refs. 12,17), whereas the phase assemblage places the weak constraint between 800 °C and 950 °C (ref. 1). Pressure estimates from projection on the haplogranitic ternary suggest <50–100 MPa, whereas Rhyolite-MELTS yields 44 ± 11 to 47 ± 32 MPa depending on oxidation state, all of which require an assumption of volatile content and fail to model this system in that they produce quartz + plagioclase ± orthopyroxene, when the observed phases are plagioclase + two clinopyroxenes[18]. The pressure of magma is often,

[1]Department of Earth and Environmental Sciences, Ludwig-Maximilians-Universität München, Munich, Germany. [2]School of Earth and Environment, University of Canterbury, Christchurch, New Zealand. ✉e-mail: J.Birnbaum@lmu.de

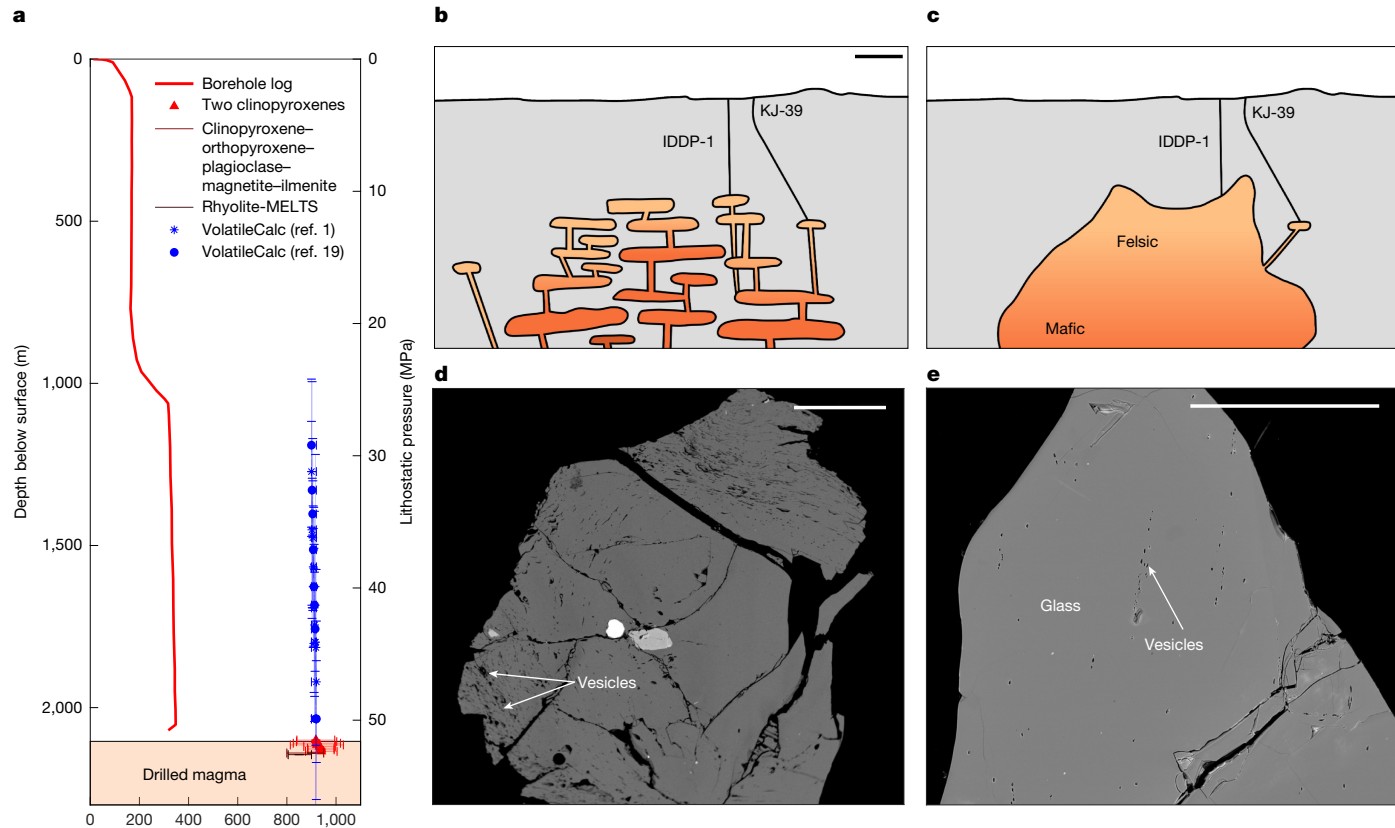

**Fig. 1 | Conflicting petrological constraints on Krafla magma storage conditions. a**, Temperature (red) with depth in the borehole[6] versus geothermometric constraints from augite–pigeonite and clinopyroxene–orthopyroxene–plagioclase–magnetite–ilmenite and Rhyolite-MELTS[1] plotted at the recovery depth, with a secondary axis showing lithostatic pressure at the corresponding depth and saturation pressure (blue, from VolatileCalc[1,19]; error bars indicate 1σ uncertainty). **b,c**, Although a trans-crustal mush arrangement of magma storage (**b**) is typically favoured[5], classic views of a basalt-underplated continuous, large rhyolite source (**c**) have also been proposed at Krafla[12,13,41] and are not distinguishable based on present geophysical observations[8–10]. **d,e**, Backscattered electron images of IDDP-1 glass chips that show the variability in vesicularity. Scale bars, 1 km (**b**); 500 μm (**d,e**).

instead, determined by the volatile ($H_2O$ and $CO_2$) concentrations of crystal-hosted melt inclusions, which are sensitive to pressure and temperature and less susceptible to decompression-induced changes than the residual melt (quenched to glass). Melt inclusions are arguably inappropriate to investigate the pressure and saturation state of the IDDP-1 magma for three reasons: (1) the few crystals present in the IDDP-1 chips frequently show dissolution (melt embayments and rounding)[1], so a meaningful, reliable population has not been isolated; (2) the pressure/depth of melt inclusions are usually determined on the basis of the assumption of volatile saturation and are, by their very nature, contained within individual crystals and thus separated from mineral pairs, which could provide an independent pressure determination; and (3) are produced preferentially during disequilibrium crystal growth and so systematically oversample non-equilibrated conditions during transport or perturbations from background[3] and therefore are poorly suited to investigate equilibrium storage. Instead, we are left with the measured volatile contents of the glass, which correspond to saturation pressures between about 35 MPa and 45 MPa (Fig. 1a and Extended Data Fig. 1), below the lithostatic pressure (about 50–57 MPa) and above the hydrostatic pressure of the well (about 16 MPa)[1]. These measurements have been interpreted to suggest that the magma is either (1) stored and degassed to equilibrium at a pressure less than lithostatic owing to interaction with the hydrothermal system[1,12] or (2) originally undersaturated[14,19].

Although the drilling fluids rapidly cooled the magma, it was still subject to decompression following intersection by the drill string[14], resulting in remobilization; the magma flowed 8 m up the well in 9 min

(ref. 20). So we ask, can the IDDP-1 glass be used as a direct record of storage conditions? Here we expand a bubble growth model for $H_2O$ (ref. 21) to include $CO_2$, coupled to a model for water species interconversion[22]. As well as the dynamics of bubble growth, a direct output of this model is the residual volatile content in the melt/glass between bubbles from which we constrain the magma storage conditions of pressure, temperature and volatile composition against the measured glass chips.

The glass fragments contain total water of 1.3–2.0 wt%, with an average of 1.8 wt%, consistent across several studies[1,11,13,19,23,24], and 50–200 ppm $CO_2$ (refs. 1,19). Vesicularity in the quenched glasses is low, with most chips having <6 vol%, although occasionally up to about 15 vol% (ref. 14). Bubble sizes are 1.5–75.0 μm, with an increase in bubble size with drilling time[14]. Bubble number densities range between $10^{11.7}$ and $10^{15}$ m$^{-3}$, which are inferred to have nucleated during drilling-induced decompression at rates of $10^5$–$10^7$ Pa s$^{-1}$ (refs. 14,25). $OH/H_2O_m$ ratios are between 1.68 ± 0.45 and 2.19 ± 0.37, increasing over time[19], and are slightly lower in more vesicular, 2.07 ± 0.20, than in less vesicular, 2.13 ± 0.38, fragments[1].

We explore which pressure–temperature ($P$–$T$) paths best reproduce the volatile chemistry and vesicularity of the IDDP-1 glass. Although the cooling from the drilling fluid should be rapid, we use a thermal model to seek paths consistent with measurements of the natural glass; geospeedometry through differential scanning calorimetry indicates that the IDDP-1 magma cooled through the glass transition regime ($T_g$) at 7–80 °C min$^{-1}$, with a $T_g$ of about 480 °C (ref. 26). We begin with one-dimensional cooling in a planar geometry from an initial temperature of 900 °C by means of conduction and forced convection at

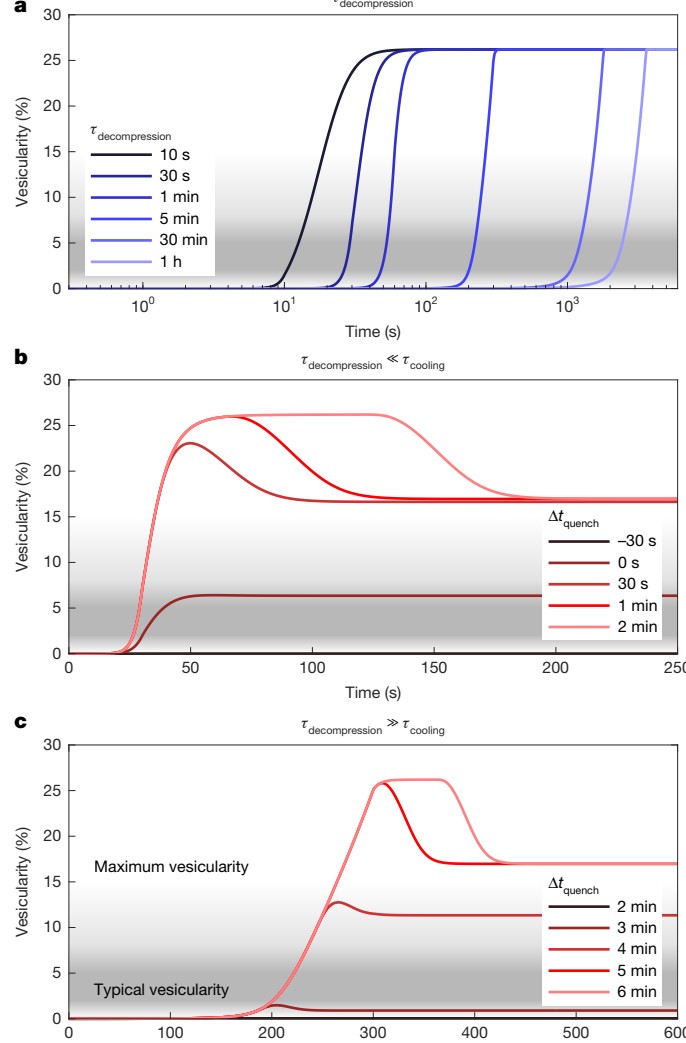

**a** $\tau_{\text{decompression}}$

$\tau_{\text{decompression}}$
- 10 s
- 30 s
- 1 min
- 5 min
- 30 min
- 1 h

**b** $\tau_{\text{decompression}} \ll \tau_{\text{cooling}}$

$\Delta t_{\text{quench}}$
- −30 s
- 0 s
- 30 s
- 1 min
- 2 min

**c** $\tau_{\text{decompression}} \gg \tau_{\text{cooling}}$

Maximum vesicularity

Typical vesicularity

$\Delta t_{\text{quench}}$
- 2 min
- 3 min
- 4 min
- 5 min
- 6 min

**Fig. 2 | Magma vesicularity evolution during different $P$–$T$–$t$ regimes requires that cooling arrests decompression. a–c**, Vesicularity along different decompression timescales (**a**) and cooling timescales along rapid (30 s) (**b**) and slow (5 min) (**c**) decompression paths. The shaded regions represent the typical ranges of vesicularity measured in the IDDP-1 glass chips, with darker shading representing greater abundance among measured chips. Line colour represents the timescales for decompression (blue) and quenching offset (red).

the surface with steam (400 °C and 16 MPa), in which drilling fluid is in direct contact with the melt. At a planar interface, cooling rates far exceed those measured in the glass. However, at larger length scales, thermal diffusion is inefficient, such that we can only reproduce the cooling rates in a narrow region between 0.3 mm and 1.3 mm from the interface (Extended Data Fig. 2). To produce a nearly homogeneous glass, similar cooling rates must be sustained across a large portion of the sampled material. This is consistent with fracturing or fragmentation of the material during cooling (Fig. 1d,e) to enhance the surface area and enable cooling from several directions.

If we assume that magma fragments (by any mechanism) during the onset or early progression of cooling, we should model the thermal evolution instead using a spherical geometry. Under these conditions, we reproduce the measured cooling rates at $T_g$ in the interior of melt fragments of radius 9–20 mm (Extended Data Fig. 3), a spacing common in perlitic glass[27,28]. This results in a timescale for cooling from storage to $T_g$ (at 480 °C) in about 4 min, with a decelerating cooling rate.

Now we turn to the pressure path experienced by the magma. During drilling, the magma should experience rapid decompression, reflected

by high bubble number densities[14,25]. However, the volatile composition and vesicularity of the glass changed only modestly over 9 h of drilling[14], suggesting that decompression of the magma was also progressive. We suggest that the decompression could be offset at increasing depth by the pressure drop generated by viscous resistance to flow up the borehole. We estimate the relevant length scale using the Hagen–Poiseuille equation:

$$\Delta P = \frac{8\mu H^2}{R^2} \frac{1}{(1+\phi)^2} \frac{\partial \phi}{\partial t} \tag{1}$$

using a borehole radius, $R$, of 0.15 m (12.5″ diameter), a melt viscosity, $\mu$, of $3.2 \times 10^5$ Pa s for magma at 900 °C and 1.8 wt% $H_2O$, a vesicularity, $\phi$, of 1–5%, bubble number density of $10^{11}$–$10^{13}$ m$^{-3}$ (ref. 25) and a rate of bubble growth of 5 μm s$^{-1}$, resulting in a $\partial \phi / \partial t \approx 1$–10% s$^{-1}$. We find a length scale, $H$, of about 1.5–4.0 m over which the pressure increases back to the investigated storage conditions (35–55 MPa). This length scale would become shorter directly proportional to an increase in viscosity. This has two implications for magma response: (1) the growth of bubbles in a partially confined space acts to buffer the pressure of magma reservoirs, such that perturbations from drilling probably only have a local (centimetres to metres) region of influence, and consequently (2) most magma may continue to experience near-storage pressures until the onset of cooling.

The model outcomes above suggest that magma conditions are modulated by cooling and decompression fronts, whose rates of propagation will be closely linked to, and driven by, fragmentation. The shear rate for magma flowing in the conduit is about 1–3 s$^{-1}$, insufficient to produce large-scale shear-driven fragmentation at this viscosity[29]. Similarly, rapid decompression can result in explosive fragmentation but requires large pressure drops for low-porosity magmas[30]. Instead, we suggest that non-explosive fragmentation from thermal shocking may be the prevalent fragmentation mechanism, similar to fuel–coolant reactions and the production of hyaloclastite and other subaqueous products[31]. We use the planar thermal simulations above and a thermal expansivity of about $6 \times 10^{-6}$ °C$^{-1}$ to find a region of high strain rate in a propagating front that moves at about 30 μm s$^{-1}$. Any crack opening would depend on the tensile strength of the material and be assisted by fluid injection and volume expansion within the cracks. This crack opening results in an 'unzipping' of the magma, leading to progressive decompression.

Because we do not simulate the exact fragmentation dynamics, we consider a range of possible decompression rates from the initial volatile saturation pressure (35–55 MPa) evolving towards hydrostatic (16 MPa) over 10 s, 30 s, 1 min, 5 min, 30 min and 1 h, corresponding to decompression rates 5.3–2,900 kPa s$^{-1}$ (Extended Data Table 1). The initial conditions are specified by choosing a starting volatile saturation pressure and molar ratio of water and $CO_2$ in the coexisting vapour phase, which together set the total water and $CO_2$ volatile contents. With a vesiculation model, we cannot distinguish between the two competing hypotheses of an undersaturated magma (which produces a first volatile phase when the system pressure drops to the saturation pressure) and magma stored saturated, but at less-than-lithostatic pressure with a negligibly small proportion of coexisting vapour. Figure 2 shows results with initially 1.8 wt% $H_2O_t$ and 111 ppm $CO_2$ at 45 MPa and 900 °C and a $OH/H_2O_m$ ratio of 4.92 (at equilibrium).

Complete decompression from 45 MPa and 900 °C and 35 MPa and 920 °C to 16 MPa produces final vesicularities of 26.2% and 25.6% and water concentrations of 1.34 wt% and 1.32 wt%, respectively (Fig. 2), both more vesicular and drier than the measured glass. At magmatic temperatures, bubble growth is rapid and only at the highest decompression rates (<5 min) is there a substantial kinetic delay in which bubble growth continues for 20–30 s after the end of decompression. This suggests that quenching of the glass must begin before the magma is completely decompressed.

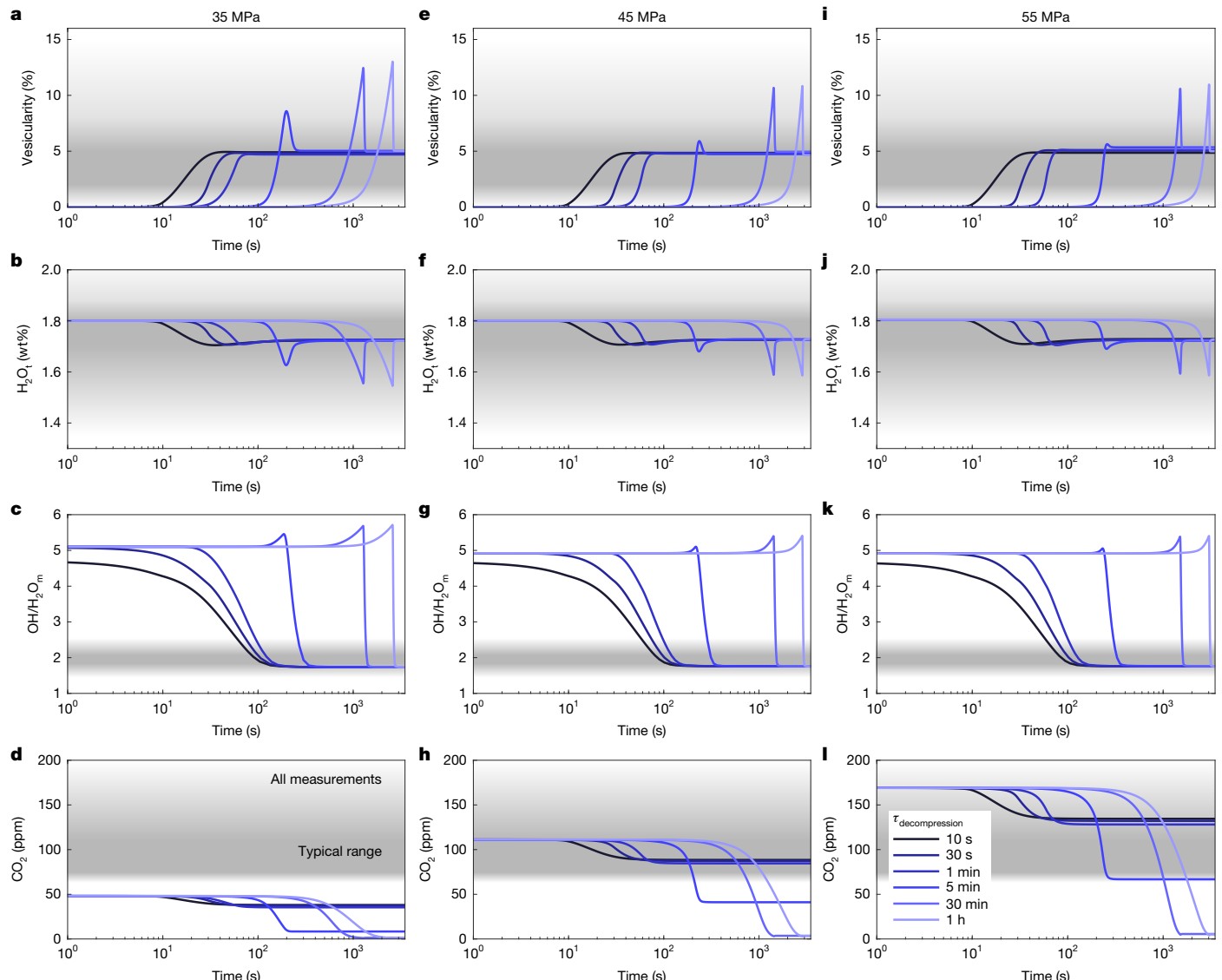

**Fig. 3 | Simulations of vesicularity need to be combined with water and $CO_2$ to distinguish different $P$–$T$ paths, which result in initial volatile-saturated, lithostatic conditions. a–l**, Vesicularity (**a**,**e**,**i**), total water (**b**,**f**,**j**), OH/$H_2O_m$ ratio (**c**,**g**,**k**) and $CO_2$ concentrations (**d**,**h**,**l**) in the melt/glass along different decompression paths from 35 MPa (**a**–**d**), 45 MPa (**e**–**h**) and 55 MPa (**i**–**l**) initial saturation pressure with syn-decompression quenching whose onset is chosen to reproduce the total vesicularity.

We simulate decompression interrupted by cooling at some time offset, $\Delta t_{quench}$. When the timescales for decompression are short compared with cooling (Fig. 2b) or when cooling occurs late, the magma nearly completely degases, but cooling produces resorption of bubbles; although decompression reduces solubility, cooling does the reverse and the decompressed magma can resorb up to approximately 10 vol% vesicles during cooling. However, to produce final vesicularities within the measured range, cooling must be early or rapid compared with decompression (Fig. 2c), in which case the final vesicularity is sensitive to the relative timing of cooling onset and the glass locks in a low vesicularity and high water content.

To find the likely storage conditions and decompression paths, we restrict the timing of cooling onset to produce the desired final vesicularity of about 5%. Because water is the volumetrically dominant species, the final water concentration of the glass and the vesicularity are closely related (Fig. 3a,b,e,f,i,j). Together they help constrain the initial water content (assuming minimal loss to the surroundings), but there remain a large number of possible $P$–$T$ paths that are suitable.

Similarly, the OH/$H_2O_m$ ratio is primarily dependent on the cooling path. Accordingly, all simulations produce a narrow range of values between 1.75 and 1.76 (Fig. 3c,g,k), which corresponds to an equilibrium temperature of roughly 500 °C. Because the cooling rate close to the glass transition was constrained by the measured glass transition temperatures and the thermal modelling, the OH/$H_2O_m$ ratio has little further discriminatory power but is consistent with the geospeedometry[26].

Conversely, owing to the large differences in water and $CO_2$ diffusivity (one to two orders of magnitude), the final $H_2O$/$CO_2$ ratio of the glass is highly path-dependent. On initial decompression, both $H_2O_m$ and $CO_2$ diffuse into the bubbles, but the slower transport of $CO_2$ increases the $H_2O$/$CO_2$ ratio in the bubble, which in turn changes the solubility of both species in the melt. This has the overall effect of delaying rapid bubble growth in the first approximately 10 MPa of decompression compared with the pure water case. On cooling, we see rapid water resorption but no marked change in the $CO_2$ dissolved in the melt, both because resorption is kinetically limited and owing to the changing $H_2O$/$CO_2$ ratio in the gas, which may allow $CO_2$ to continue moving into the bubble. As a result, the $CO_2$ in the final glass is more sensitive to the initial vesiculation than to resorption.

Because the $CO_2$ has preferentially been lost to the bubbles, previous estimates using a mixed solubility model and only the glass chemistry[1]

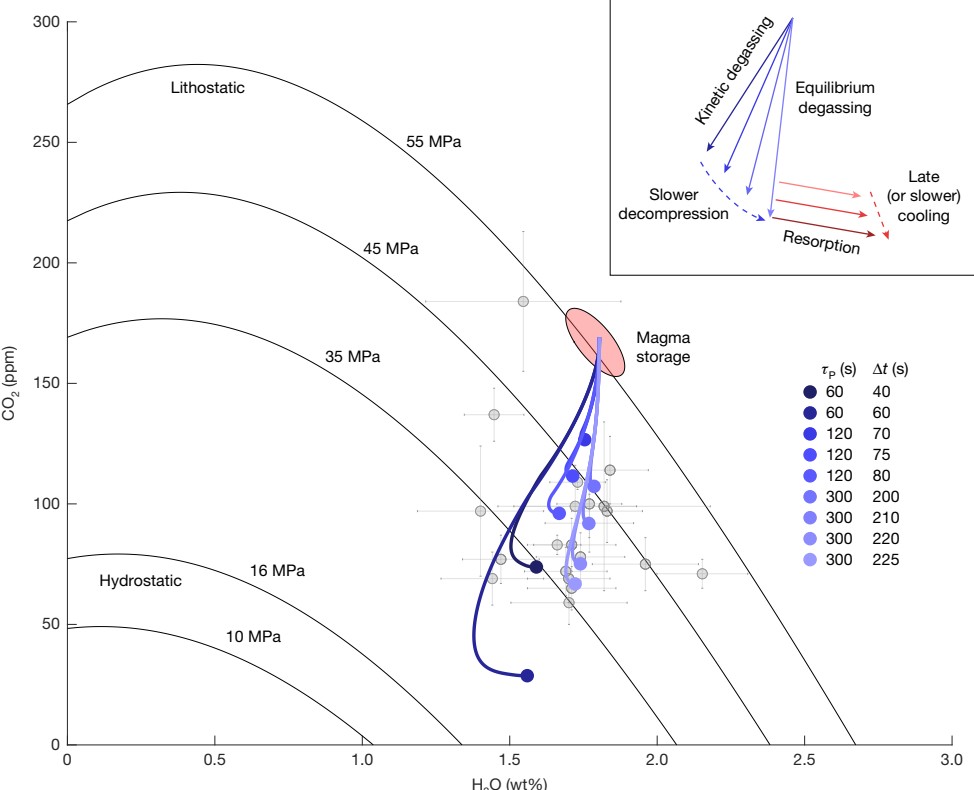

**Fig. 4 | Small variations in decompression timescales between 1 min and 5 min with synchronous cooling cover the range of measured glass volatiles.** Calculated paths (blue lines) of average $H_2O$ and $CO_2$ in the melt along decompression from magma storage (sketched in red) at lithostatic pressure to the final glass (blue markers), which span the field of measured glasses[1,19] (grey markers; error bars indicate 1$\sigma$ uncertainty). Black lines show the saturation pressure[42] at 900 °C. The inset highlights the trajectory of the volatile saturation path during decompression at different rates and synchronous resorption.

would have resulted in an underestimate of storage saturation pressure. Although we can produce reasonable glass chemistry along rapid decompression and early cooling from 45 MPa, decompression from lithostatic pressure (55 MPa) using a modestly higher initial $CO_2$ concentration of 170 ppm and 1.8 wt% water better encompasses the range of measured volatile concentrations in the glass chips (Fig. 4). We cannot produce any glasses with more than about 50 ppm $CO_2$ from 35 MPa. Starting from a cooler (about 850 °C) initial temperature would have only a modest effect on starting $CO_2$ (121 versus 108 ppm at 45 MPa and 1.77 wt% $H_2O$) and so this method cannot place tighter constraints on the starting temperature compared with existing geothermometers and experimental studies[32].

To preserve sufficient $CO_2$ in the melt, we require that the characteristic timescale for decompression must be $\lesssim$5–10 min, similar to the characteristic timescale for cooling, and cooling onset must be approximately synchronous with or slightly trailing ($\lesssim$3–4 min) decompression (Figs. 4 and 5). If decompression occurs over the 1.5–4.0 m characteristic length scale found above, we find propagation rates of 15–80 cm s$^{-1}$, resulting in a fragmentation level that is nearly stationary or rises up the borehole with the flow of magma. The magma cannot have undergone slow decompression for tens of minutes to hours during approach of the drill string, suggesting that the roof rock is capable of isolating the magma from the surrounding hydrothermal system, at least on short timescales.

In our simulations, diffusion is rapid enough to almost completely erase the gradient in both total water content and speciation except in the very last stages of cooling while $CO_2$ either minimally resorbs or even continues to diffuse into the bubble (Extended Data Fig. 4). This suggests that resorption may be both pervasive[33–35] and difficult to detect in the final glass, in which a notable gradient towards the bubble is conventionally used as the primary indicator of resorption[35–37].

This also has implications for our interpretation of the origin of the magmatic volatiles; the $\delta D$, $\delta^{18}O$ and total water contents have been interpreted to evidence mixing between mid-ocean ridge basalts and meteoric water, along with crustal assimilation by assuming that the isotopic signatures are not substantially altered by exsolution owing

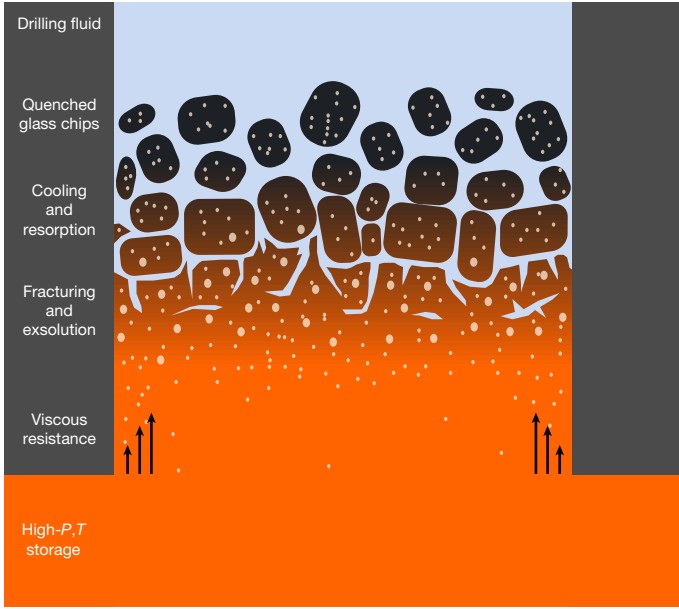

**Fig. 5 | Decompression and cooling occur synchronously during thermal quench fragmentation.** Sketch of processes during magma perturbation by drilling.

to the low final vesicularity[14]. However, vesiculation followed by rapid rate-limited resorption contributes to fractionation, in which the heavier isotopes are enriched in the bubble during exsolution, and cannot diffuse as rapidly during resorption[38–40].

Our study highlights the rapid, disequilibrium changes that overprint equilibrium conditions and the importance of carefully accounting for the magma physico-chemical response during quenching; we demonstrate that glass is not a direct representation of storage, even under rapid cooling. During drilling, magma at Krafla responded to pressure and temperature perturbations over extremely short timescales (seconds to minutes) and small length scales (centimetres to metres), closely tied to fragmentation driven by thermal shock. By correcting for this rapid response using our multiparametric model, we produce the first, to our knowledge, robust constraint of magmatic conditions, confirming storage of volatile-saturated magma at lithostatic pressure. The ability to invert for magmatic conditions using drilling samples during rapid cooling and quenching at depth, however, raises the challenge inherent in reconstructing storage conditions from volcanic products that have traversed kilometres of crust over hours to weeks. Beyond the validation of magmatic conditions, the thermo-physico-chemico-rheological-kinetic model developed here provides a first rigorous tool to engineer magma response and enable safe access through drilling by forward modelling to find the optimal drilling conditions (for example, borehole geometry, penetration rate, fluid composition) to inhibit magma ascent up the borehole.

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

# Methods

### Thermal modelling

Cooling of the magma was induced by interaction with drilling fluid. Although we expect cooling rates to vary in space and time, we seek thermal paths that are consistent with measurements of the natural glass.

We consider cooling in one dimension in either planar

$$\rho c_p \frac{\partial T}{\partial t} = \frac{\partial}{\partial x}\left(k\frac{\partial T}{\partial x}\right) \tag{2}$$

or radially symmetric geometry

$$\rho c_p \frac{\partial T}{\partial t} = \frac{1}{r^2}\frac{\partial}{\partial r}\left(r^2 k\frac{\partial T}{\partial r}\right) \tag{3}$$

in which we choose a melt density, $\rho$, of 2,380 kg m$^{-3}$, heat capacity, $c_p$, which is temperature and water content dependent[43–45] and a function of vesicularity[46], and a thermal conductivity, $k$, which has a melt thermal conductivity of 1.3364 W m$^{-1}$ °C$^{-1}$ and can also vary depending on vesicularity[46].

We choose the boundary conditions to be no flux at some increasing depth, $L$, or at the centre of radial symmetry and a flux condition at the magma–drilling fluid interface. We simplify the flux by assuming that the dominant heat transport mechanism at the interface is forced convection:

$$Q = h(T_{\text{rhyolite}} - T_{\text{fluid}}) \tag{4}$$

with a heat transfer coefficient, $h$, of 6,000 W m$^{-2}$ °C$^{-1}$, and a constant fluid temperature, $T_{\text{fluid}}$, of 400 °C (ref. 11), consistent with a thin film of boiling water.

We solve the thermal evolution using the finite difference method on a three-point stencil and use the backwards differentiation formula 2 (BDF2):

$$T_n = \frac{4}{3}T_{n-1} - \frac{1}{3}T_{n-2} + \frac{2}{3}\delta t f(T_n) \tag{5}$$

and bootstrap in to the method using the backward differentiation formula BDF1 (backward Euler method).

### Hagen–Poiseuille scaling

We begin with the customary formulation:

$$\Delta P = \frac{8\mu H Q}{\pi R^4} \tag{6}$$

in which $\Delta P$ is the pressure drop, $\mu$ is the viscosity, $H$ is the characteristic length along the flow direction, $R$ is the conduit radius and we need to calculate the volumetric flux, $Q$, from known variables of the system. Neglecting buoyancy-driven flow, the volumetric flux comes only from the expansion of the gas phase, which has a volume of:

$$V_g = 4/3\pi a^3 N_b (\pi R^2 H) \tag{7}$$

for a bubble radius, $a$, bubble number density, $N_b$, conduit radius, $R$, and characteristic height of $H$. Because

$$\phi = \frac{V_g}{V_m + V_g} \tag{8a}$$

$$V_m = \pi R^2 H - 4/3\pi a_0^3 N_b(\pi R^2 H) \tag{8b}$$

and assuming an initially negligible bubble volume fraction, we can rewrite the rate of change of gas volume:

$$\frac{\partial V_g}{\partial t} = \pi R^2 H \frac{\partial}{\partial t}\left(\frac{\phi}{1-\phi}\right) = \pi R^2 H \frac{\partial}{\partial t}\left(\frac{1}{(1-\phi^2)}\right)\frac{\partial \phi}{\partial t} \tag{9}$$

and substituting into equation (6):

$$\Delta P = \frac{8\mu H^2}{R^2}\left(\frac{1}{(1-\phi^2)}\right)\frac{\partial \phi}{\partial t} \tag{10}$$

we arrive at the approximate pressure drop from bubble-growth-driven expansion of the material up a cylindrical conduit.

### Thermal shock fragmentation

We investigate whether cooling induces fragmentation in the near vicinity of the melt–drilling fluid interface (1 mm) over short duration (1 s) using the thermal model above. The viscosity of the melt at each location and time is determined by the temperature and composition[47]. We assume a strain rate in tension:

$$\dot{\gamma} = \alpha\frac{\partial T}{\partial t} \tag{11}$$

for a linear thermal expansivity, $\alpha = 6 \times 10^{-6}$ m °C$^{-1}$ (ref. 45).

### Water content modelling

We build on the work of refs. 21,22 to numerically model the transport of $CO_2$ and transport and reaction of molecular water and hydroxyl dissolved in the glass in response to changing pressure and temperature conditions. We use the $CO_2$ diffusivity model of ref. 48 and the water diffusivity model of ref. 22, in which only molecular water is able to diffuse, and the diffusivity is a function of both total water and water speciation. We modify the total water conservation equation of ref. 21 to include diffusion of only molecular water:

$$\frac{\partial H_2O_m}{\partial t} = \frac{1}{r^2}\frac{\partial}{\partial t}\left(r^2 D \frac{\partial H_2O_m}{\partial r}\right) \tag{12}$$

in which $D$ is the diffusivity and $H_2O_m$ is the concentration of molecular water, given by:

$$H_2O_m = \frac{W_{H_2O}}{W}X_{H_2O_m} \tag{13a}$$

$$X_{H_2O_m} + \frac{1}{2}X_{OH} = 1 \tag{13b}$$

in which $W_{H_2O}$ is the molecular weight of water, $W$ is the molecular weight of the hydrous silicate melt, $X_{H_2O_m}$ is the molar fraction of molecular water and $X_{OH}$ is the molar fraction of hydroxyl. We solve for conservation of moles of hydroxyl using the reaction equation of ref. 22:

$$\frac{\partial OH}{\partial t} = -2k\left(\frac{OH^2}{K} - H_2O_m X_O\right) \tag{14a}$$

$$X_O = 1 - H_2O_m - OH \tag{14b}$$

in which $k$ is the kinetic reaction rate, $K$ is the equilibrium constant and $X_O$ is the molar fraction of oxygen.

The spherical melt shell is discretized radially with an inner boundary located at the vapour bubble wall and an outer boundary determined by the initial bubble number density and vesicularity. At the inner boundary in contact with the vapour bubble, we set the molecular water concentration according to the equilibrium total water concentration:

$$H_2O_{m,eq} = \begin{cases} H_2O_{t,eq} - OH & OH < H_2O_{t,eq} \\ 1 \times 10^{-15} & OH \geq H_2O_{t,eq} \end{cases} \quad (15)$$

The outer boundary has a symmetry (no-flux) condition. We choose as the initial state pressures between 35 MPa and 55 MPa and temperature between 900 °C and 920 °C (ref. 1). We choose initial water contents of 1.8 wt% and adjust the starting molar ratio of water and $CO_2$ in the coexisting vapour phase to reach saturation using the mixed solubility model[42]. The model is also sensitive to the assumed bubble number density, which we choose to be $3 \times 10^{14} \, m^{-3}$ (ref. 14).

## Data availability

The simulation results presented in this study are publicly available from Code Ocean at https://doi.org/10.24433/CO.9780368.v1 and Zenodo at https://doi.org/10.5281/zenodo.18741118 (ref. 49). Source data are provided with this paper.

## Code availability

The code developed in this manuscript is publicly available from Code Ocean at https://doi.org/10.24433/CO.9780368.v1.

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

**Acknowledgements** This study was financially supported by the European Research Council (ERC): Magma Outgassing During Eruptions and Geothermal Exploration (MODERATE, grant no. 101001065). We thank C. Annen, J. Hammer and B. Scaillet for their constructive reviews that improved this contribution.

**Author contributions** J.B.: conceptualization, investigation, methodology, software, visualization, writing – original draft preparation, review and editing. F.B.W.: conceptualization, methodology, writing – review and editing. J.E.K.: conceptualization, funding acquisition, supervision, writing – review and editing. B.K.: writing, review and editing. P.A.W.: conceptualization, writing – review and editing. M.M.d.S.: writing – review and editing. K.-U.H.: writing – review and editing. Y.L.: conceptualization, funding acquisition, methodology, supervision, writing – review and editing.

**Funding** Open access funding provided by Ludwig-Maximilians-Universität München.

**Competing interests** The authors declare no competing interests.

**Additional information**
**Correspondence and requests for materials** should be addressed to Janine Birnbaum.

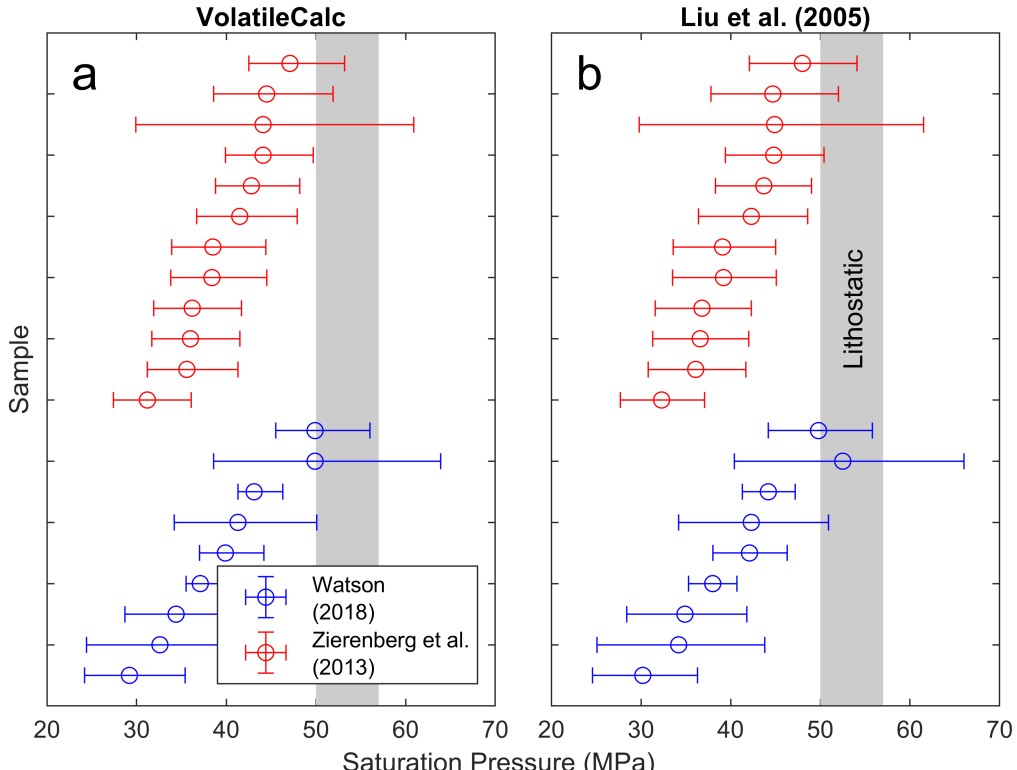

**Extended Data Fig. 1 | Remaining volatiles in the quenched glass are below lithostatic solubility.** Volatile saturation pressures from individual Fourier transform infrared measurements from Watson[19] (2018) and Zierenberg et al.[1] (red) using VolatileCalc (**a**) and Liu et al.[42] (**b**) saturation models.

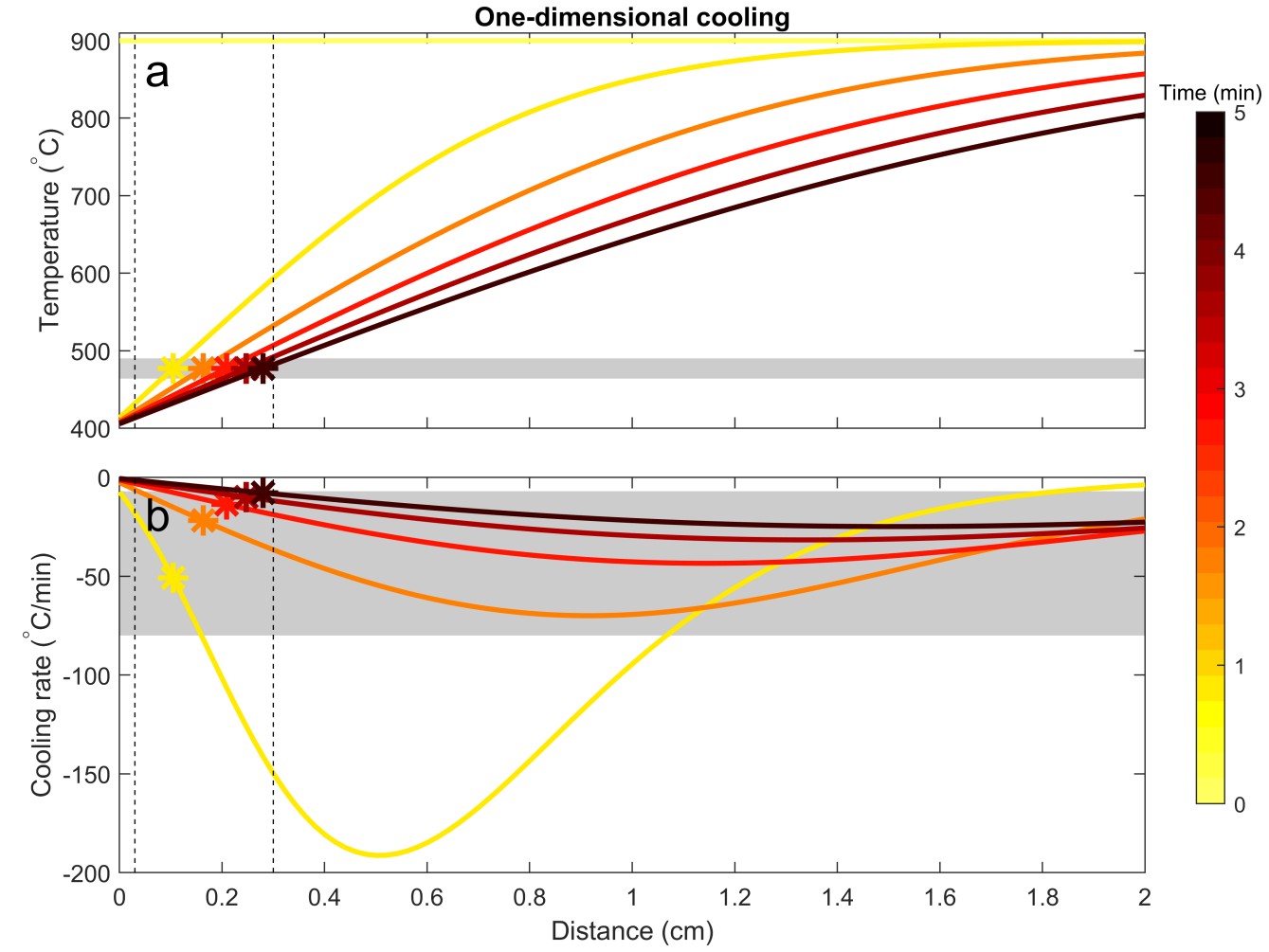

**Extended Data Fig. 2 | Cooling profile in one direction (planar).** Temperature (°C) (**a**) and cooling rate (°C min⁻¹) (**b**) coloured by time. Grey-shaded region shows the approximate glass transition temperature and stars indicate the time of the glass transition. Cooling rates match natural samples only at distances of 0.03 and 0.3 cm and timescales between 1 and 5 min.

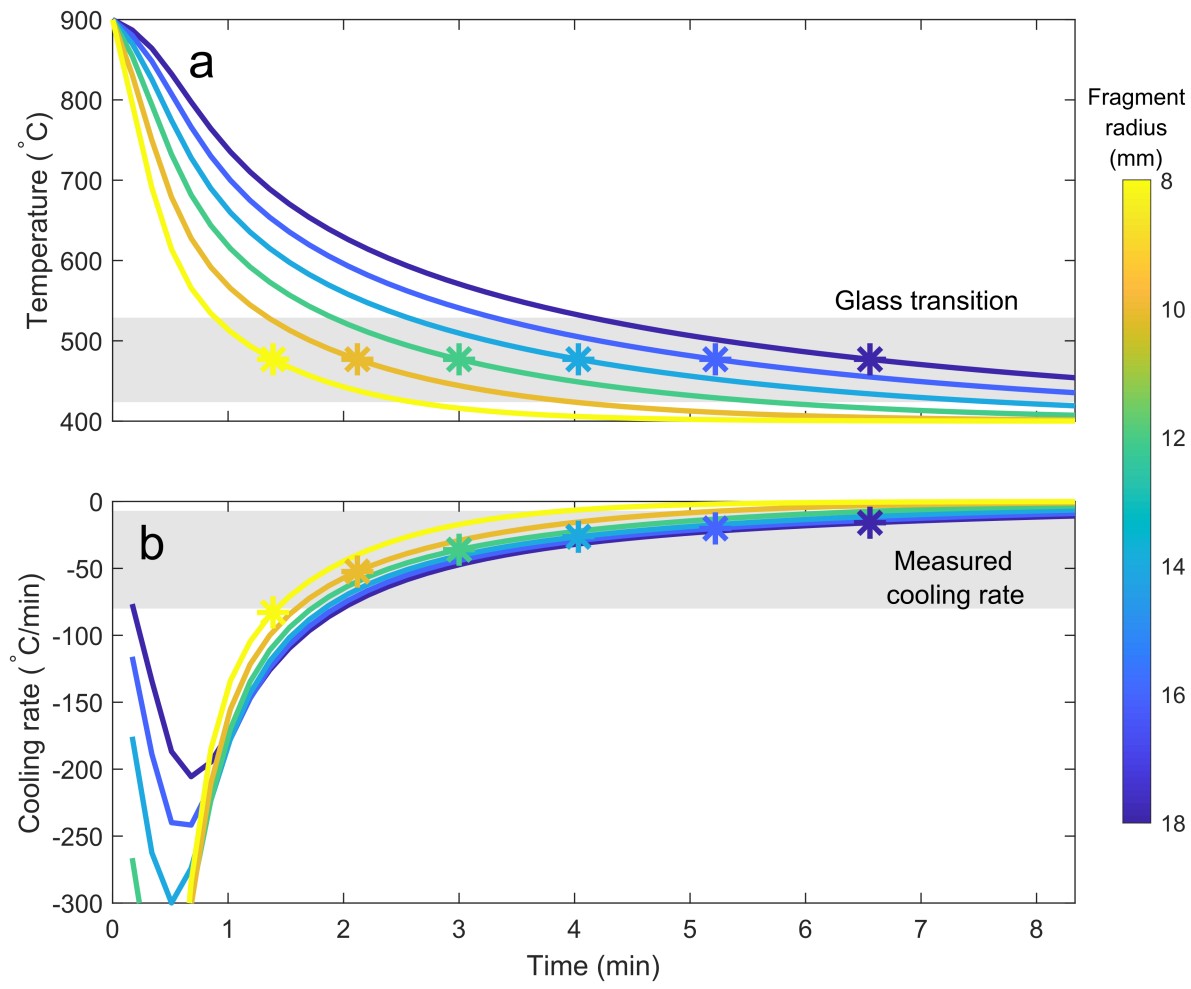

**Extended Data Fig. 3 | Cooling rate in spherical fragments.** Mean temperature (**a**) and cooling rate (**b**) for melt fragments of various radii. Grey-shaded region shows the approximate glass transition temperature and stars indicate the time of the glass transition. To reproduce the observed glass transition temperatures, fragment size must be between 9 and 20 mm radius.

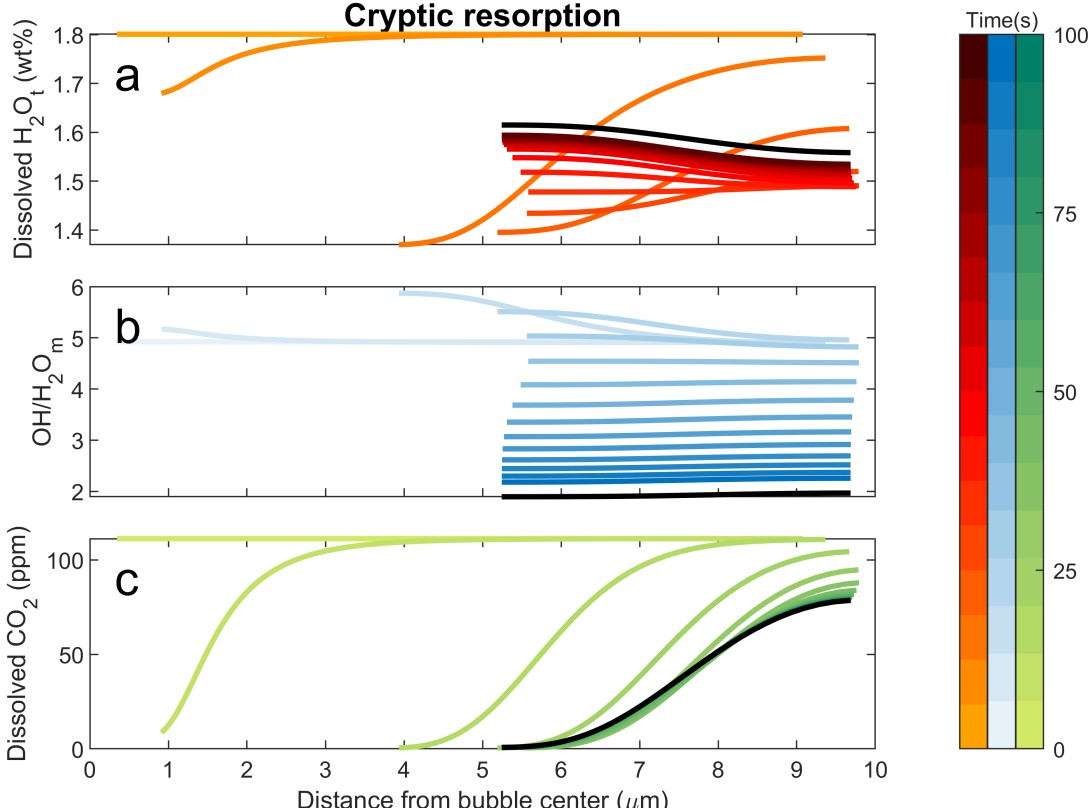

**Extended Data Fig. 4 | Volatile concentrations in quenched glass surrounding bubbles do not preserve strong evidence of resorption.** Total water (**a**), OH/$H_2O_m$ ratio (**b**) and dissolved $CO_2$ (**c**) in the melt/glass with radial distance away from the bubble centre, with lighter colours indicating earlier times. Decompression and cooling begin at $t = 0$ s and resorption becomes apparent at $t \approx 15$ s, but by the time the melt quenches at 100 s, $CO_2$ still shows a minimum towards the bubble, total water shows a very small maximum towards the bubble, probably within measurement uncertainty for, for example, Fourier transform infrared under most circumstances, and no substantial variation in water speciation with distance from the bubble.

## Extended Data Table 1 | Model input parameters

| Initial pressure (MPa) | Initial temperature (°C) | Initial $H_2O$ in the melt (wt%) | Initial $CO_2$ in the melt (ppm) | Vapor composition (Mole fraction $H_2O/H_2O+CO_2$) | Decompression rates (kPa/s) | Cooling time offset (s) |
|---|---|---|---|---|---|---|
| 35 | 920 | 1.80 | 48 | 0.78 | 5.3-1900 | -22-2590 |
| 45 | 900 | 1.80 | 110 | 0.60 | 8.1-2900 | -18-2850 |
| 55 | 900 | 1.80 | 169 | 0.50 | 11-3900 | -18-3020 |

Input saturation pressure, temperature and volatile contents with paired decompression rates (all saturation pressures are modelled with the same decompression timescales) and cooling offset times, which are constrained to produce simulations with the target final vesicularity (5 vol%) and water content (1.77 wt%).