## [Peer Review File · Nature]

Disequilibrium response to tapping crustal magma reveals storage conditions

Corresponding Author: Dr Janine Birnbaum

Version 0:

Reviewer comments:

Referee #1

(Remarks to the Author)

This is an interesting paper that looks at the magmatic rocks recovered from a drill experiment in Iceland during which a magma pocket was intercepted. The authors look at the volatile contents of matrix glass and model its evolution in an effort to shed light on storage conditions (which in this specific context are known, notably the pressure). In detail the authors use preserved water and CO₂ contents, as well as water speciation in matrix glass. Along with thermal constraints, they simulate the kinetics of volatile exsolution/resorption, including the H₂O speciation. They conclude that preserved volatile contents and speciation in volcanic matrix glass cannot be used to compute directly storage conditions. This goal can be achieved using, they say, their scheme of recalculation.

While the procedure the authors use is fine, I don't believe it holds the potential they claim to become a powerful tool for restoring storage conditions elsewhere. I rather think it is a highly specific approach. I say that for the following reasons:

1. First of all, to the best of my knowledge, no one has ever claimed that the volatile content of matrix glass can be used directly to restore storage conditions. This important goal is best achieved using phase equilibrium arguments and volatile contents preserved in melt inclusions as shown by many previous studies. One good example has been just published by Harmon et al (2025, EPSL) on the Krafla volcano, but there are many others (see papers by Rutherford and colleagues for instance). The lack of reference to this well established approach is surprising, given the context of that paper.
2. Second the authors deal with a very specific environment in which the magma is quenched during drilling by the incoming flow of water associated to drilling. This is certainly not a common situation, and I hardly see how it can be generalised to volcanoes everywhere. The question of significance of dissolved volatiles in matrix glass is an important one and the work done here shows clearly that even with a small decompression step (a few meters), the matrix glass rapidly loses storage conditions. But I can't see how the proposed scheme can be used to restore storage conditions with some confidence for magma parcels that have travelled several km upward (i.e. 8 km which is a common storage pressure of evolved magma in the crust). What the authors propose may be applicable to future drilling experiments, which are unlikely to become that common, given the associated costs. For that reason I believe this interesting work is more suited to a specialised journal.
3. On lines 48-50 it is stated that volatile saturation pressures (35-45 MPa) are significantly lower than lithostatic pressure (50-56 MPa). Calculating the volatile saturation pressure using 1.8 wt% H₂O and 170 ppm CO₂ in melt, which are well within the range of reported volatile contents, I get 56 MPa (with Liu et al model), so I don't understand this point, which is presented as evidence of our current inability to constrain storage pressure. On the contrary, to me there is perfect agreement between lithostatic and volatile saturation pressures and there is absolutely no conflict in petrological constraints, unlike what the title of figure 1 says. On figure 1 it seems that pressures of volatile saturation come from a volatilcalc model (which previous works have used) while the authors seem to use Liu et al model in their calculation. If true, this is inconsistent and should be corrected.

4. This is more a comment: on the first introductory paragraph it is stated that geothermobarometry comes along with large uncertainties, in particular pressure (± 200 MPa). While such an error may be true for Cpx-based methods, which are quite commonly used to infer storage pressure, it is not true for phase equilibrium approaches, as stated above. Taking as an example Mt St Helens eruption, I think we can say with some confidence that the magma reservoir pressure was around 200 ± 50 MPa. That is to say that in many instances, restored storage pressures of silicic reservoirs are known to better than ± 200 MPa.

5. For understandable reasons, the focus of the paper is more on pressure than on temperature, because pressure is admittedly a more difficult parameter to assess. Yet, the reported spread in pre-eruptive temperature is quite large (70°C). If at equilibrium, such a spread should come along with varying crystallinity (for constant bulk rock composition) which I understand is not observed (true?). Alternatively, besides model errors, this spread may simply reflect magma mixing, as is common in many volcanic areas. There is nothing wrong with this but in my opinion it introduces an additional source of uncertainty, given the quite strong temperature dependence of H₂O solubility in silicic melts. I am not sure (but I may have missed this point) that the modelling performed takes this explicitly into account?

6. In the abstract, it is stated that disequilibrium crystallisation during ascent complicates the unravelling of storage conditions. Again here, a word of caution is in need: while this is true for slowly ascending silicic melts, in those ascending fast, ie Plinian eruptions, very little happens from the viewpoint of crystallisation, and the preserved crystals (phenocrysts along with their melt inclusions) can be used to restore storage conditions, or any pre-eruptive phenomenon (mixing etc..). So this statement is incorrect or belongs to a specific class of eruptions, which should be termed to avoid confusion or misunderstanding from untrained readers.

7. It is unclear to me how CO₂ diffusion is handled in the model of H₂O diffusion described in the SI. The authors state that CO₂ diffuses slowly compared to H₂O, which is true for H₂O-rich conditions but no more at low water contents (< 1 wt% H₂O) where both species have similar diffusivities, CO₂ diffusing even faster than H₂O at very low H₂O content (which equation do they use? I think it is from the compilation of Zhang et al when looking at their codes but it would be better to mention that explicitly in the SI as well). It may be that this will have a marginal effect of model outcomes, but I suggest this point be better explained and perhaps elaborated upon.

(Remarks on code availability)

Referee #2

(Remarks to the Author)

1. Significance

This study makes a significant contribution by leveraging direct sampling from a borehole, an approach uniquely suited to constraining sub-volcanic systems. As the authors rightly note, such systems often remain poorly resolved by geophysical methods. The use of physical samples enables insights into the magmatic plumbing system that would otherwise remain speculative. Extracting meaningful signals from these samples requires sophisticated modeling approaches to disentangle the effects of natural processes from those introduced during acquisition and recovery. The authors' effort to meet this challenge is hampered by lack of support for a key assumption: fluid saturation of the magma at the storage/recovery depth.

2. Validity of Methods, Quality of Data, and Quality of Presentation

The authors employ a suite of plausible and technically sophisticated models, including H–O speciation, H₂O+CO₂ fluid solubility, and viscous flow dynamics, to interpret the volatile content of the recovered glass.

Minor weaknesses: (a) the application of two-clinopyroxene thermobarometry is too imprecise to resolve pressure differences over the small depth range in question. More broadly, any barometric approach relying on clinopyroxene is likely to be similarly limited in precision under these conditions. (b) Regarding the data presentation, the BSE images confirm the presence of fresh glass and two mineral phases, but they do not confirm the stability of the mineral assemblages invoked in the thermobarometric models.

3. Appropriate Treatment of Uncertainties

The manuscript would benefit from acknowledging that volatile saturation calculations provide only minimum melt storage pressure. The calculated volatile saturation pressure of 35 MPa is a lower bound; if the magma were not fluid saturated, then the actual storage pressure could have been significantly higher. Critically, the manuscript does not present compelling evidence that the magma was indeed fluid saturated at the storage depth, which is a foundational assumption of the decompression modeling. Demonstrating fluid saturation typically requires independent evidence, such as the presence of melt inclusions containing sufficient volatile concentrations to require saturation, or intact fluid bubbles within inclusions that have not been altered or decrepitated.

4. Validity of Conclusions

If the magma was fluid saturated at the lithostatic pressure (e.g., ~ 52 MPa), the presence of lower volatile concentrations (corresponding to 35 MPa saturation pressure) in the glass would necessitate thoughtful modeling of degassing and

resorption processes, as pursued in this exercise. However, in the absence of firm evidence for fluid saturation at the outset, the necessity of such modeling disappears. That is, if the magma was undersaturated at the time of recovery, there would be no need for dynamic pressure 'buffering' during ascent to cause reabsorption of volatiles. This alternate assumption, volatile undersaturation, emerges as a plausible and perhaps even more consistent explanation for the observations than the authors' original premise of fluid saturation. This is itself a noteworthy result, given the challenges associated with critically determining fluid saturation state in natural systems. If the authors reframe their findings accordingly, the impact of the study is still compelling.

I acknowledge that I have not reviewed all of the references cited in the manuscript about these drill cores, and it is possible that my concerns are allayed therein. Nonetheless, I hope the comments provided here will be helpful for at least highlighting areas of potential confusion that other readers may also encounter.

5. Suggested Improvements

Several aspects of the manuscript would benefit from clarification and revision, regardless of the fluid saturation assumption. The attached PDF includes many comments and suggestions for improvement, keyed to lines of text and figures.

6. Clarity

The manuscript's clarity would improve with more consistent treatment of key variables. The selection and use of pressure values are sometimes ambiguous—at times treated as constraints derived from observations (e.g., lines 48–49), and at other times as model variables or assumptions. A similar issue arises with temperature, which is alternately described as a measured parameter (e.g., lines 78, 81) and a modeled outcome (e.g., line 79). These inconsistencies make it difficult for the reader to follow the logic of the analysis.

(Remarks on code availability)

I encountered a few difficulties executing the reviewer copy of the code, which I documented in screenshots.

1. Some things are a little unclear (how to specify location of results, as directed in the instructions when user clicks "run").
2. The only output I saw was the set of "t = " values at each iteration written to the window; output did not include plots, and instead there were error codes.
3. Message in window "no preview available (the file is too large) gave the options of downloading a file or opening in a new window. Both options prompted me to download a .mat file, which I can open in Matlab (of course), but is not illuminating with respect to output.

Version 2:

Reviewer comments:

Referee #1

(Remarks to the Author)

I have carefully read the revised version of this paper, and the accompanying rebuttal letter, and found that the authors have correctly answered the questions raised during the first round of review, clarifying the most important issues I could see. I thank them for their efforts in explaining some aspects in a very detailed way. I am happy to recommend acceptance of this revised version.

I have made a few minor additional comments/remarks reported below, and keyed to the line number of the manuscript.

52-53. Note that the « weak » constraints on temperature are very specific to this Icelandic system, but cannot be taken as representative of T estimates elsewhere. Again quoting Mt St Helens or Pinatubo, which represent a kind of canonical studies, temperature constraints gained from petrology are quite accurate

57-58. That is correct.

62-70. A word of caution/comment here. It maybe that melt inclusions are inappropriate, or cannot be found in abundance, at Iceland, but they were quite appropriate at many other volcanic centers to set tight constraints on pre-eruptive storage conditions, in particular when combined with phase equilibrium considerations. Saying that all melt inclusions result from disequilibrium is an extreme statement : some of them may indeed grow during decompression, but some other just grow in the reservoir and are likely to record storage conditions (for instance the melt inclusion with the highest CO₂ content in figure 4). I also point out that in several instances melt inclusions do occur along with fluid inclusions, and this is taken as evidence of fluid saturation, in which case the H₂O and CO₂ systematics gained from melt inclusions analyses correctly inform about P (and not just on minimum of P storage. As written now, the text may give (to readers unfamiliar with this theme) the wrong impression that crystals in magmas everywhere cannot be trusted, and that only the matrix glass is likely to inform about storage. I would slightly reword this section by making it clear that this (restriction) is specific to the case study.

67-68. Assuming that some melt inclusions were trapped during ascent is equivalent to say that some crystals grew during ascent. But are the time scales of magma ascent and crystal growth comparable ?

122. Something important to take into consideration here is that the magma will flow upward using the fracture path made by the drilling itself. In nature, magma has to break-up frontal rocks itself and this will consume energy as well as slow the process.

130-131 ; A sensitivity analysis on the role of melt viscosity would be useful here.

141-142. Please give a reference for this threshold of shear driven fragmentation. Perhaps Spieler et al. 2004 ?

241-243. This is an important conclusion in particular for felsic systems. The high viscosity of rhyolites is often taken as evidence for their enhanced potential (relative to mafic melts) to preserve HT processes which is clearly not the case.

259. ...during rapid cooling and quenching at depth.

440. This is a PhD thesis. Is it available to a large audience ?

448. This also looks like a PhD reference but is multi authored ? The same remark applies as to the availability of that work.

(Remarks on code availability)

I did not try to run the code (this is hardly a plug and play exercise). However the explanations about the physical and chemical principles behind the code are very well laid out in the method section and I dont think they require complex numerical schemes.

Referee #3

(Remarks to the Author)

The manuscript uses data from the drill hole that sampled Krafla's magma and calculates with numerical models the temperature, pressure, volatile content, and vesicularity evolution of the magma upon drilling.

I am reviewing the revised version and not the original version. This version addresses well the concerns raised by the first round of review. In particular, it clarifies that volatile saturation is not an assumption but a result. It also clarifies that this work is important because it shows in a quantitative way the limits of determining pressure of magma residence based on phase equilibria, melt inclusions, or other thermobarometers.

Detailed comments:

The authors mention the possibility of a magma stored at a less than lithostatic pressure. Could they add a sentence to specify under what circumstances the pressure can be below the lithostatic pressure.

l.79-82: sentence starting with "Here,". I do not understand the sentence, especially where it says "[..] changes in the residual melt/glass between bubbles from which[..]" I suggest to rephrase. Maybe it would be clearer with two sentences.

l. 268: I suggest to be more explicit on how the study produces a tool to engineer magma response and enable safe access via drilling.

Extended data figure 2: Please, specify in the caption if the cooling is planar or radial.

Catherine Annen

(Remarks on code availability)

Because of time limitation, I did not looked at the code in detail, but I read the readme file. It could be clearer by specifying what the different scripts and functions are doing (main.m, Numerical_Model_spec_v2_CO2.m, getFunctions_v2.m, getFunctions_speciation.m). The functions refer to a user manual but I could not see it. I also noticed that readme mentions main.py but the code is main.m.

Referee #1:

This is an interesting paper that looks at the magmatic rocks recovered from a drill experiment in Iceland during which a magma pocket was intercepted. The authors look at the volatile contents of matrix glass and model its evolution in an effort to shed light on storage conditions (which in this specific context are known, notably the pressure). In detail the authors use preserved water and CO₂ contents, as well as water speciation in matrix glass. Along with thermal constraints, they simulate the kinetics of volatile exsolution/resorption, including the H₂O speciation. They conclude that preserved volatile contents and speciation in volcanic matrix glass cannot be used to compute directly storage conditions. This goal can be achieved using, they say, their scheme of recalculation.

While the procedure the authors use is fine, I don't believe it holds the potential they claim to become a powerful tool for restoring storage conditions elsewhere. I rather think it is a highly specific approach. I say that for the following reasons:

1. First of all, to the best of my knowledge, no one has ever claimed that the volatile content of matrix glass can be used directly to restore storage conditions. This important goal is best achieved using phase equilibrium arguments and volatile contents preserved in melt inclusions as shown by many previous studies. One good example has been just published by Harmon et al (2025, EPSL) on the Krafla volcano, but there are many others (see papers by Rutherford and colleagues for instance). The lack of reference to this well established approach is surprising, given the context of that paper.

While we agree with the reviewer that the volatile content of the matrix glass is not representative of the storage conditions, measurements of total volatile content and isotopic ratios have been used in precisely this manner at Krafla (Elders, 2011; Zierenberg et al., 2013; Watson, 2018; Seligman et al. 2019; Saubin et al., 2021), and also for quench pressures under ice or water (e.g., Tuffen et al. 2010 and references therein in which the authors discuss equilibrium and non-equilibrium degassing during ascent, but not during quenching).

Naturally, the Harmon et al. (2025) paper is an important piece to discuss that has been published since our initial submission. Even in this recent paper, they find minor disagreement between the calculated (1.6-1.9 km) and known depths (2.1 km) – a 10-20% error, which highlights the uncertainty in these methods and they are still unable to reconcile the long-standing issue of model predictions in which orthopyroxene and quartz are predicted but not observed in the samples, and clinopyroxene is not predicted, but is observed. Because most of these analyses rely on the same databases of experiments and often the same models, other works come to essentially the same conclusions (Zierenberg et al., 2013; Masotta et al. 2018).

However, the critical point in our paper is that analyses via melt major element chemistry and phase equilibria do not tell us about volatile saturation state, and in fact usually require an assumption of it (Harmon et al., 2025 use the Zierenberg et al., 2013 glass volatile compositions directly). Ideally, a study such as ours should be complimentary to analysis of phase equilibria or other crystal geochemistry, and one does not replace the other because they are posed to answer different questions about the system.

Experimental work on the IDDP-1 glasses have shown an ability to reproduce equilibrium assemblages matching the observed crystal cargoes under certain fO₂ conditions (Masotta et

al., 2018, da Silva et al., in prep), but these studies have restored only the H₂O and not the CO₂ contents to account for volatile loss, to which these experiments are sensitive.

Melt inclusions pose a further challenge. The depths (pressures) of melt inclusions are typically inferred on the basis of volatile saturation, which 1) ignores the possibility of disequilibrium conditions which are likely present to promote the rapid crystal growth that results in trapped melt, and 2) consequently, melt inclusion populations systematically over-sample disequilibrium transport and perturbations to storage conditions, and potentially miss long-term storage. Not to mention that melt inclusions are themselves subject to a variety of post-entrapment processes including water loss through the crystal and the growth of crystals or vapor shrinkage bubbles inside the melt inclusions and can include co-entrapped vapor bubbles. Like phase equilibria above, melt inclusion analyses serve a specific purpose, but are poorly posed to answer the question we tackle in this work.

In this specific case there are also practical limitations to the use of melt inclusions; the few crystals present in the IDDP-1 glass cuttings are poor in melt inclusions and a representative number (dozens or hundreds for good analysis) have not been isolated.

We are happy to expand our discussion of this topic given constraints on the length of the text, and certainly to include the new work of Harmon et al. (2025), but we feel strongly that the tool we present can answer a different question than these established methods.

2. Second the authors deal with a very specific environment in which the magma is quenched during drilling by the incoming flow of water associated to drilling. This is certainly not a common situation, and I hardly see how it can be generalised to volcanoes everywhere. The question of significance of dissolved volatiles in matrix glass is an important one and the work done here shows clearly that even with a small decompression step (a few meters), the matrix glass rapidly loses storage conditions. But I can't see how the proposed scheme can be used to restore storage conditions with some confidence for magma parcels that have travelled several km upward (i.e. 8 km which is a common storage pressure of evolved magma in the crust). What the authors propose may be applicable to future drilling experiments, which are unlikely to become that common, given the associated costs. For that reason I believe this interesting work is more suited to a specialised journal.

We appreciate that drilling into magma is an unusual circumstance, although one that we believe will be of increasing frequency and importance as we search for higher energy density resources to meet global needs for heat and electricity. We concur that our work highlights the challenge in attempting to reconstruct the storage conditions of natural eruptions and make this point on lines 218-221. However, we feel that the insights we gain into non-eruptive magma storage conditions in this work will be of widespread interest and address a question not normally possible from either natural eruptive products or exposed extinct systems.

3. On lines 48-50 it is stated that volatile saturation pressures (35-45 MPa) are significantly lower than lithostatic pressure (50-56 MPa). Calculating the volatile saturation pressure using 1.8 wt% H₂O and 170 ppm CO₂ in melt, which are well within the range of reported volatile contents, I get 56 MPa (with Liu et al model), so I don't understand this point, which is presented as evidence of our current inability to constrain storage pressure. On the contrary, to me there is perfect agreement between lithostatic and volatile saturation pressures and there is absolutely no conflict in petrological constraints, unlike what the title of figure 1 says. On figure 1 it seems that pressures of volatile saturation come from volatilcalc model (which previous works have

used) while the authors seem to use Liu et al model in their calculation. If true, this is inconsistent and should be corrected.

The difference between VolatileCalc and the Liu et al. (2005) model are not large enough to account for this difference. These calculations have been done on the basis of individual H₂O-CO₂ pairs on the same glass chip, rather than selecting a global maximum for both H₂O and CO₂ separately. You can see that one sample from Watson (2018) pushes into the lithostatic region, but has a particularly high uncertainty. On the whole, these samples suggest a low likelihood of representing saturated conditions without corrections for degassing and resorption. And, I would reiterate that the published interpretation has been that of undersaturation.

4. This is more a comment: on the first introductory paragraph it is stated that geothermobarometry comes along with large uncertainties, in particular pressure (± 200 MPa). While such an error may be true for Cpx-based methods, which are quite commonly used to infer storage pressure, it is not true for phase equilibrium approaches, as stated above. Taking as an example Mt St Helens eruption, I think we can say with some confidence that the magma reservoir pressure was around 200 ± 50 MPa. That is to say that in many instances, restored storage pressures of silicic reservoirs are known to better than ± 200 MPa.

We are reporting here the value from Wieser et al. (2025) which is indeed for clinopyroxene. We can expand this point to include other methods, but for our purposes here ± 50 MPa is still a large uncertainty (25%) and not sufficient to resolve shallow storage.

5. For understandable reasons, the focus of the paper is more on pressure than on temperature, because pressure is admittedly a more difficult parameter to assess. Yet, the reported spread in pre-eruptive temperature is quite large (70°C). If at equilibrium, such a spread should come along with varying crystallinity (for constant bulk rock composition) which I understand is not observed (true?). Alternatively, besides model errors, this spread may simply reflect magma mixing, as is common in many volcanic areas. There is nothing wrong with this but in my opinion

it introduces an additional source of uncertainty, given the quite strong temperature dependence of H₂O solubility in silicic melts. I am not sure (but I may have missed this point) that the modelling performed takes this explicitly into account?

Perhaps this was unclear in the text. The 70 °C window is an uncertainty on the temperature calculation on the basis of petrological constraints, not representative of a spread across the samples. Also, the crystals represent such a small volume fraction of the magma that many of the small glass chips are aphyric, so it would not be possible to search for systematic differences in crystallinity across chips. The chips don't show large variability in major or trace element that would suggest unequilibrated magma mixing. We chose a smaller range of temperature to present in the paper (900-920 °C) that were consistent with experiments conducted on this composition which become too crystal rich at lower temperature (and require higher temperatures at lower pressures, meaning drier initial conditions). But, the sensitivity to temperature in this range is much lower than the sensitivity to pressure, so for 45 MPa at 900 °C (result we present) the starting volatile budget at saturation is 1.77 wt% H₂O and 108 ppm CO₂, and at 850 °C for the same water content (which can't vary much and still match the observed vesicularity and remaining water in the glass), CO₂ only goes up to 121 ppm. Not enough to make a difference in the conclusions. We would be happy to perform more full simulations at these conditions if required, but expect a small effect on the results.

6. In the abstract, it is stated that disequilibrium crystallisation during ascent complicates the unravelling of storage conditions. Again here, a word of caution is in need: while this is true for slowly ascending silicic melts, in those ascending fast, ie Plinian eruptions, very little happens from the viewpoint of crystallisation, and the preserved crystals (phenocrysts along with their melt inclusions) can be used to restore storage conditions, or any pre-eruptive phenomenon (mixing etc.). So this statement is incorrect or belongs to a specific class of eruptions, which should be termed to avoid confusion or misunderstanding from untrained readers.

This is a broad summary that includes both crystallization and vesiculation which matters for inverting volatile concentrations in even fast-ascending eruptions.

7. It is unclear to me how CO₂ diffusion is handled in the model of H₂O diffusion described in the SI. The authors state that CO₂ diffuses slowly compared to H₂O, which is true for H₂O-rich conditions but no more at low water contents (< 1 wt% H₂O) where both species have similar diffusivities, CO₂ diffusing even faster than H₂O at very low H₂O content (which equation do they use? I think it is from the compilation of Zhang et al when looking at their codes but it would be better to mention that explicitly in the SI as well). It may be that this will have a marginal effect of model outcomes, but I suggest this point be better explained and perhaps elaborated upon.

We never reach low water contents, the lowest water contents we consider are still above 1.5 wt% H₂O. We indeed use the model of Zhang et al. (2007), we apologize if this was unclear.

Referee #2:

1. Significance

This study makes a significant contribution by leveraging direct sampling from a borehole, an approach uniquely suited to constraining sub-volcanic systems. As the authors rightly note, such

systems often remain poorly resolved by geophysical methods. The use of physical samples enables insights into the magmatic plumbing system that would otherwise remain speculative. Extracting meaningful signals from these samples requires sophisticated modeling approaches to disentangle the effects of natural processes from those introduced during acquisition and recovery. The authors' effort to meet this challenge is hampered by lack of support for a key assumption: fluid saturation of the magma at the storage/recovery depth.

We apologize if this was in some way unclear. The purpose of this paper is to investigate the saturation pressure, we don't assume fluid saturation, but rather it is the result of the paper. By simulating decompression from 35 and 45 MPa (below lithostatic pressure) we are directly looking at the effect of the melt being undersaturated. We describe the results in terms of the equilibrium pressure to which that volatile load corresponds for convenience. Using this method we cannot distinguish between an undersaturated magma and saturated magma that is being stored under-pressured, but we find that neither scenario matches the observations. The "assumption" of equilibrium with a particular pressure in theory places an additional constraint on the H₂O-CO₂ ratio in the melt and vapor phases, but because we find that H₂O is tightly constrained by the final melt and vary the vapor phase composition (H₂O-CO₂ ratio in the melt) to match the final water, we are really only varying the CO₂ content which would be the free variable without this assumption.

2. Validity of Methods, Quality of Data, and Quality of Presentation

The authors employ a suite of plausible and technically sophisticated models, including H–O speciation, H₂O+CO₂ fluid solubility, and viscous flow dynamics, to interpret the volatile content of the recovered glass.

Minor weaknesses: (a) the application of two-clinopyroxene thermobarometry is too imprecise to resolve pressure differences over the small depth range in question. More broadly, any barometric approach relying on clinopyroxene is likely to be similarly limited in precision under these conditions. (b) Regarding the data presentation, the BSE images confirm the presence of fresh glass and two mineral phases, but they do not confirm the stability of the mineral assemblages invoked in the thermobarometric models.

We give the thermobarometric study for context in the existing literature. We agree that the constraint it imposes has a large uncertainty, but our real pressure constraint comes from the known depth of the magma chamber. Otherwise, we investigate the effect of pressure in the simulations. We refer the reviewer to the existing literature on the mineral phases present in the IDDP-1 magma.

3. Appropriate Treatment of Uncertainties

The manuscript would benefit from acknowledging that volatile saturation calculations provide only minimum melt storage pressure. The calculated volatile saturation pressure of 35 MPa is a lower bound; if the magma were not fluid saturated, then the actual storage pressure could have been significantly higher. Critically, the manuscript does not present compelling evidence that the magma was indeed fluid saturated at the storage depth, which is a foundational assumption of the decompression modeling. Demonstrating fluid saturation typically requires independent evidence, such as the presence of melt inclusions containing sufficient volatile concentrations to require saturation, or intact fluid bubbles within inclusions that have not been altered or decrepitated.

Our results are compatible only with a scenario in which the magma is both saturated and stored at lithostatic pressure as either case of 1) volatile undersaturation for a magma stored at lithostatic pressure or 2) a magma stored under-pressured are both inconsistent with the final measured glass chemistries after correcting for the response of magma to drilling. We haven't assumed this, but rather directly asked whether either of these scenarios can explain the data. There are not sufficient melt inclusions present in the small crystal load to investigate volatile contents from earlier in the magma history, but we would argue the presence of the high bubble number density found in the recovered chips indicates that the magma cannot have been far from saturation or syn-decompression cooling would have prevented vesiculation.

4. Validity of Conclusions

If the magma was fluid saturated at the lithostatic pressure (e.g., ~52 MPa), the presence of lower volatile concentrations (corresponding to 35 MPa saturation pressure) in the glass would necessitate thoughtful modeling of degassing and resorption processes, as pursued in this exercise. However, in the absence of firm evidence for fluid saturation at the outset, the necessity of such modeling disappears. That is, if the magma was undersaturated at the time of recovery, there would be no need for dynamic pressure 'buffering' during ascent to cause reabsorption of volatiles. This alternate assumption, volatile undersaturation, emerges as a plausible and perhaps even more consistent explanation for the observations than the authors' original premise of fluid saturation. This is itself a noteworthy result, given the challenges associated with critically determining fluid saturation state in natural systems. If the authors reframe their findings accordingly, the impact of the study is still compelling.

In addition to above, we apologize if the discussion around resorption was unclear. We hypothesize that the resorption is triggered by the change in solubility of the melt in response to cooling, not the pressure changes. The observation of a relatively homogeneous population of glass chips with drilling time over several hours and bubble number densities indicative of rapid decompression rates requires that the magma pressure be maintained away from the drill string. A volatile undersaturated magma would have insufficient CO₂ to match the joint constraints of vesicularity, and water and CO₂ contents of the final glass.

I acknowledge that I have not reviewed all of the references cited in the manuscript about these drill cores, and it is possible that my concerns are allayed therein. Nonetheless, I hope the comments provided here will be helpful for at least highlighting areas of potential confusion that other readers may also encounter.

5. Suggested Improvements

Several aspects of the manuscript would benefit from clarification and revision, regardless of the fluid saturation assumption. The attached PDF includes many comments and suggestions for improvement, keyed to lines of text and figures.

We thank the reviewer for the detailed comments on the text. We present a subset of comments here that require a response (beyond minor text or graphics edits) and are not otherwise addressed here:

Figure 1 shows the lithostatic pressure at the recovery depth is ~52 MPa.

We plot the temperature estimates of the magma on the depth y-axis in red because they were recovered from that depth. The pressure estimates in blue all correspond to the secondary y-axis and show pressures up to ~30-50 MPa. We can clarify this in the caption.

“be more specific about this pressure modeling based on volatile saturation. what H₂O and CO₂ and temperature combinations yield this range of pressures?”

This comment seems to be asking about the compiled data from Zierenberg et al. (2013) and Watson (2018) which we can add to the supplement. We will also add a table of the model input conditions for clarity.

“is the rock at 2000 m both porous and water saturated?”

We give the hydrostatic pressure because that is certainly the pressure in the well. Generally, our results are consistent with at least the rock directly overlying the magma (felsite with small degree of partial melt) to be impermeable.

“It's hard to tell whether the voids in panel c are vesicles or just pores introduced during preparation of the surface. The glass chip in panel e doesn't seem to have any vesicles. The emphasis on bubbles at this point in the text is hard to reconcile with the image data.”

There are some fractures in panel d, but we interpret the small voids to be vesicles (~12%). We would argue that panel e shows a small fraction of vesicles (~0.5%). We show here the full range of vesicle textures that are present in the chips, but emphasize that throughout the manuscript we are discussing a typical vesicularity of only 5%.

“should be possible to infer temperature from the speciation for known total H₂O?”

Speciation is highly sensitive to the cooling path and happens rapidly enough to keep pace with the cooling unless 1) the cooling is extremely rapid (only achievable in laboratory conditions) or 2) the diffusion and reaction rate become very slow as the melt viscosity increases which in our case happens about 100 °C above the glass transition temperature. We show results for this.

Lines 76-88 regarding the geospeedometry constraints

This constraint comes from geospeedometry results from Wadsworth et al. (2024) which has no explicit modeling in it. Differential scanning calorimetry was performed that allows for determination of the glass transition temperature which depends on both melt chemistry and cooling rate. An initial measurement is performed on the natural glass and then the sample is repeatedly heated and cooled across the glass transition at different rates. The glass transition temperature of the initial sample can then be compared to known cooling/heating rates which reveals the cooling rate of the sample. We can clarify this in the text.

seems like the presence of fragments would be corroborated by the state of the material recovered (shard shapes), unless this is obscured by the drilling process.... clarify.

There are some large glass shards recovered, and some which show evidence of interaction with water over the whole surface, but any quantitative estimate of the size distribution of the fragments would indeed be overprinted by secondary drill action.

6. Clarity

The manuscript's clarity would improve with more consistent treatment of key variables. The

selection and use of pressure values are sometimes ambiguous—at times treated as constraints derived from observations (e.g., lines 48–49), and at other times as model variables or assumptions. A similar issue arises with temperature, which is alternately described as a measured parameter (e.g., lines 78, 81) and a modeled outcome (e.g., line 79). These inconsistencies make it difficult for the reader to follow the logic of the analysis.

We use the pre-existing constraints to select an appropriate range of pressures to test with the numerical simulations, we can clarify this in the text. See response to reviewer 1 above regarding the selection and effects of temperature.

Referee #2 (Remarks on code availability):

I encountered a few difficulties executing the reviewer copy of the code, which I documented in screenshots.

We apologize if the reviewer had trouble running the code.

1. Some things are a little unclear (how to specify location of results, as directed in the instructions when user clicks "run").

The Code Ocean service requires the results to be written to "results", which the code is set up to do and then saves under a title string that includes the run parameters. We can add a readme with further documentation. Otherwise, the messages that appear asking about the results location can be ignored, it will save to that directory.

2. The only output I saw was the set of "t = " values at each iteration written to the window; output did not include plots, and instead there were error codes.

This is correct, the only outputs to the command line is a progress update on the simulation time and there are some warnings that come only from the plotting and can be safely ignored. We can suppress them in the code. It looks to have run correctly in the attached screenshots.

3. Message in window "no preview available (the file is too large) gave the options of downloading a file or opening in a new window. Both options prompted me to download a .mat file, which I can open in Matlab (of course), but is not illuminating with respect to output.

That file contains the simulation results. We can add the code with the simulation visualization to the code repository.

Referee comments are in black, responses are shown in blue, and all line numbers refer to the revised manuscript without tracked changes.

Referee #1:

This is an interesting paper that looks at the magmatic rocks recovered from a drill experiment in Iceland during which a magma pocket was intercepted. The authors look at the volatile contents of matrix glass and model its evolution in an effort to shed light on storage conditions (which in this specific context are known, notably the pressure). In detail the authors use preserved water and CO₂ contents, as well as water speciation in matrix glass. Along with thermal constraints, they simulate the kinetics of volatile exsolution/resorption, including the H₂O speciation. They conclude that preserved volatile contents and speciation in volcanic matrix glass cannot be used to compute directly storage conditions. This goal can be achieved using, they say, their scheme of recalculation.

We thank the reviewer for their appraisal of our study and their helpful comments that have allowed us to improve the manuscript.

While the procedure the authors use is fine, I don't believe it holds the potential they claim to become a powerful tool for restoring storage conditions elsewhere. I rather think it is a highly specific approach. I say that for the following reasons:

1. First of all, to the best of my knowledge, no one has ever claimed that the volatile content of matrix glass can be used directly to restore storage conditions. This important goal is best achieved using phase equilibrium arguments and volatile contents preserved in melt inclusions as shown by many previous studies. One good example has been just published by Harmon et al (2025, EPSL) on the Krafla volcano, but there are many others (see papers by Rutherford and colleagues for instance). The lack of reference to this well established approach is surprising, given the context of that paper.

While we agree with the reviewer that the volatile content of the matrix glass is not representative of the storage conditions, measurements of total volatile content and isotopic ratios have been used in precisely this manner for quench pressures under ice or water (e.g., Tuffen et al. 2010: <https://doi.org/10.1016/j.earscirev.2010.01.001> and references therein in which the authors discuss equilibrium and non-equilibrium degassing during ascent, but not during quenching), for “pre-explosive” stagnation pressures (e.g., Burgisser et al. 2020: <https://doi.org/10.1016/j.jvolgeores.2020.107072> and references therein), and at Krafla (Elders, 2011; Zierenberg et al., 2013; Watson, 2018; Seligman et al. 2019; Saubin et al., 2021; and indeed Harmon et al., 2025), where the measurements have been used to interpret undersaturated conditions. Hence, we believe that our approach is justified by the need for new integrated, multiparametric tools to reconcile quenched-in volatile species in groundmass glass to the vesicular network. In the presence of such integrated datasets, one needs to question what the volatile signature means about the state of magma; accordingly, we provide the following changes to bolster the importance of resolving the cause for observed volatile concentrations.

The following text was added to line 73: “These measurements have been interpreted to suggest that the magma is either 1) stored and degassed to equilibrium at a pressure less than lithostatic^[1,12] or 2) originally undersaturated^[14,16].”

Naturally, the Harmon et al. (2025) paper is an important piece to discuss that has been published since our initial submission. Even in this recent paper, they find minor disagreement between the calculated (1.6-1.9 km) and known depths (2.1 km) – a 10-20% error, which highlights the uncertainty in these methods (i.e., an uncertainty far too large to discriminate between lithostatic, sub-lithostatic, or even super-lithostatic conditions), and they are still unable to reconcile the long-standing issue of model predictions in which orthopyroxene and quartz are predicted but not observed in the samples, and clinopyroxene is not predicted, but is observed. Because most of these analyses rely on the same databases of experiments and often the same models, other works come to essentially the same conclusions (Zierenberg et al., 2013; Masotta et al. 2018) – an important shortcoming of these methods when applied to chemically-different magmatic systems.

Experimental work on the IDDP-1 glasses have shown an ability to reproduce equilibrium assemblages matching the observed crystal cargoes under certain fO_2 conditions (Masotta et al., 2018, da Silva et al., in prep), but these studies have restored only the H_2O and not the CO_2 contents to account for volatile loss, to which these experiments are sensitive.

The following text was added to line 54: “Pressure estimates from projection on the haplogranitic ternary suggest <50-100 MPa, while Rhyolite-MELTS yields 44 ± 11 MPa to 47 ± 32 MPa depending on oxidation state, all of which require an assumption of volatile content, and fail to model this system in that they produce quartz + plagioclase \pm orthopyroxene, when the observed phases are plagioclase + two clinopyroxenes¹⁸.”

However, the critical point in our paper is that analyses via melt major element chemistry and phase equilibria do not tell us about volatile saturation state, and in fact usually require an assumption of it (Harmon et al., 2025 use the Zierenberg et al., 2013 glass volatile compositions directly). Ideally, a study such as ours should be complimentary to analysis of phase equilibria or other crystal geochemistry, and one does not replace the other because they are posed to answer different questions about the system. We have made these two competing hypotheses clearer in the lines mentioned above.

Melt inclusions pose a further challenge. The depths (pressures) of melt inclusions are typically inferred on the basis of volatile saturation, which 1) ignores the possibility of disequilibrium conditions which are likely present to promote the rapid crystal growth that results in trapped melt, and 2) consequently, melt inclusion populations systematically over-sample disequilibrium transport and perturbations to storage conditions, and potentially miss long-term storage. Not to mention that melt inclusions are themselves subject to a variety of post-entrapment processes including water loss through the crystal and the growth of crystals or vapor shrinkage bubbles inside the melt inclusions and can include co-entrapped vapor bubbles. Like phase equilibria above, melt inclusion analyses serve a specific purpose, but are poorly posed to answer the question we tackle in this work.

In this specific case there are also practical limitations to the use of melt inclusions; the few crystals present in the IDDP-1 glass cuttings are poor in melt inclusions and a representative number (dozens or hundreds for good analysis) have not been isolated.

The tool we present is posed to answer a different question than these established methods.

The following text has been added to line 58: “The pressure of magma is often, instead, determined by the volatile (H_2O and CO_2) concentrations of crystal-hosted melt inclusions which are sensitive to pressure and temperature and less susceptible to decompression-induced

changes than the residual melt (quenched to glass). Melt inclusions are arguably inappropriate to investigate the pressure and saturation state of the IDDP-1 magma for three reasons: 1) the few crystals present in the IDDP-1 chips frequently show dissolution (melt embayments and rounding)¹, so a meaningful, reliable population has not been isolated, 2) the pressure/depth of melt inclusions are usually determined on the basis of the assumption of volatile saturation and are, by their very nature, separated from mineral pairs which could provide an independent pressure determination, and 3) are produced during disequilibrium crystal growth, so systematically over-sample non-equilibrated conditions during transport or perturbations from background³, and therefore cannot investigate equilibrium storage.”

2. Second the authors deal with a very specific environment in which the magma is quenched during drilling by the incoming flow of water associated to drilling. This is certainly not a common situation, and I hardly see how it can be generalised to volcanoes everywhere. The question of significance of dissolved volatiles in matrix glass is an important one and the work done here shows clearly that even with a small decompression step (a few meters), the matrix glass rapidly loses storage conditions. But I can't see how the proposed scheme can be used to restore storage conditions with some confidence for magma parcels that have travelled several km upward (i.e. 8 km which is a common storage pressure of evolved magma in the crust). What the authors propose may be applicable to future drilling experiments, which are unlikely to become that common, given the associated costs. For that reason I believe this interesting work is more suited to a specialised journal.

We appreciate that drilling into magma is, as yet, an unusual circumstance, although one that we believe will be of increasing frequency and importance as we search for higher energy density resources to meet global needs for heat and electricity. However, this rare case gives us access and insight to a ubiquitous process that usually remains out-of-reach. Decompression and cooling during transport from storage conditions are always present in volcanic samples, and we show that even short duration (<5 min) and little relative movement (<10 m) result in important, disequilibrium changes in the magma. We concur that our work highlights the challenge in attempting to reconstruct the storage conditions of natural eruptions that travel kilometers through the crust, and make this point on lines 218-221 of the original manuscript (now lines 263-266). If volcanic transport processes are so efficient at overprinting the initial storage condition, then we should exploit this relatively tightly constrained system to test hypotheses about magma storage that are untestable in volcanic samples, and add caution to the interpretation of such samples using equilibrium models.

As we mention in our response to point 1 above, volatiles in glasses quenched in lava, especially when erupted under ice or water, are often interpreted to reflect the last pressure of equilibrium experienced by those lavas. We show that water-cooled quench at 16 MPa (~2 km water depth), appropriate to many submarine and subglacial conditions, occurs synchronously with decompression degassing and can result in rapid water resorption which leads to underestimates of the storage pressure. This model is applicable to help us understand decompression- and cooling-induced changes in, for example, pillow lavas, where inversion for pressure and temperature during effusion may be useful. Even without the simplifying assumption of recent equilibration, kinetic models of degassing and resorption could lead to future understanding of the volatile transport processes in obsidians more broadly that often show water and CO₂ concentrations (e.g. Rust & Cashman, 2007:

<https://doi.org/10.1007/s00445-006-0111-4>), water speciation (e.g. Castro et al., 2014:

<https://doi.org/10.1016/j.epsl.2014.08.012>), and syn-eruptive crystallization (Castro & Mercer,

2004: <https://doi.org/10.1029/2004GL020489>) inconsistent with equilibrium at their emplacement conditions. Meaningful progress on understanding the formation of obsidian during explosive eruptions will require mechanistic explanations for outgassing and gas transport, and potentially resorption, for which this model lays the groundwork.

Our work provides the first robust evidence for the oft-debated, long-held claim that magma may sit at lithostatic conditions – an assertion which has hitherto been unvalidated. The insights we gain into non-eruptive magma storage conditions in this work (i.e. that magma is stored at lithostatic pressure and volatile saturated) will be of widespread interest and address a question not normally possible from either natural eruptive products or exposed extinct systems. Most volcanoes spend the majority of their time in repose, with magma stored and evolving in the crust. As a community, we don't understand well what fraction of magma escapes from the crust and what process dominate the magma that remains trapped; our understanding of this must naturally be limited if we only make observations on the magmas that do erupt, or remnant fossil systems that have crystallized, degassed, been uplifted and eroded. Volcanic eruptions are the product of perturbations to these typical storage conditions, and as a result their products are often used to search for eruption triggers, that is, the perturbations to the system that destabilize storage. Accurate conceptual models of magmatic/volcanic systems and interpretations of volcanic precursory signals require an understanding of storage at a detail that has not been possible with existing methods. This begins with a better understanding of this system at Krafla and at other drilling experiments, which provide us with insights not accessible through existing methods or volcanic products, and in aggregate can improve our picture of the subsurface of volcanoes.

As above, we have revised the manuscript to better clarify the existing competing hypotheses that we rule out and add justification on why we believe this tool is important to exploit this rich measurement.

3. On lines 48-50 it is stated that volatile saturation pressures (35-45 MPa) are significantly lower than lithostatic pressure (50-56 MPa). Calculating the volatile saturation pressure using 1.8 wt% H₂O and 170 ppm CO₂ in melt, which are well within the range of reported volatile contents, I get 56 MPa (with Liu et al model), so I don't understand this point, which is presented as evidence of our current inability to constrain storage pressure. On the contrary, to me there is perfect agreement between lithostatic and volatile saturation pressures and there is absolutely no conflict in petrological constraints, unlike what the title of figure 1 says. On figure 1 it seems that pressures of volatile saturation come from volatilecalc model (which previous works have used) while the authors seem to use Liu et al model in their calculation. If true, this is inconsistent and should be corrected.

The differences between VolatileCalc and the Liu et al. (2005) model are not large enough to account for this difference (typically 1-5% difference). These calculations have been done on the basis of individual H₂O-CO₂ pairs on the same glass chip, rather than selecting a global maximum for both H₂O and CO₂ separately. You can see that one sample from Watson (2018) pushes into the lithostatic region, but has a particularly high uncertainty. On the whole, these samples suggest a low likelihood of representing lithostatic saturated conditions without corrections for degassing and resorption. And, we would reiterate that the published interpretation has been that of undersaturation.

The following figure has been added to the extended data:

Extended data figure 1: Volatile saturation pressures from individual FTIR measurements from Watson (2018, blue) and Zierenberg et al. (2013, red) using A) VolatileCalc and B) Liu et al. (2005) saturation models.

Additionally, we have added the following figure to the main text of the paper which we hope makes clearer where the individual glass measurements sit in on a typical H₂O-CO₂ diagram with contours of saturation pressure for 900 °C as well as the range of plausible pressure-temperature-time paths that all start from lithostatic storage and produce final glasses that resemble the measurements.

Figure 4: Small variations in decompression timescales between 1 and 5 minutes with synchronous cooling cover the range of measured glass volatiles. Calculated paths (solid blue lines) of average H₂O and CO₂ in the melt along decompression from magma storage (sketched

in red) at lithostatic pressure to the final glass (blue markers) which span the field of measured glasses^{1,23} (gray markers). Black lines show the saturation pressure¹⁷ at 900 °C. The inset schematic highlights the trajectory of the volatile saturation path during decompression at different rates and synchronous resorption.

4. This is more a comment: on the first introductory paragraph it is stated that geothermobarometry comes along with large uncertainties, in particular pressure (± 200 MPa). While such an error may be true for Cpx-based methods, which are quite commonly used to infer storage pressure, it is not true for phase equilibrium approaches, as stated above. Taking as an example Mt St Helens eruption, I think we can say with some confidence that the magma reservoir pressure was around 200 ± 50 MPa. That is to say that in many instances, restored storage pressures of silicic reservoirs are known to better than ± 200 MPa.

We are reporting here the value from Wieser et al. (2025) which is indeed for clinopyroxene. We expanded this point to include other methods, but for our purposes here ± 50 MPa is still an uncertainty as large as the total pressure, and not sufficient to resolve shallow storage.

Lines 25-27 were edited to read: "Geothermobarometers using mineral-mineral chemistry or phase equilibria have been applied to volcanic materials to constrain magmatic origins, but these methods have large uncertainties (~ 50 -200 MPa or ~ 2.5 -10 km)."

The pressure estimates for this system from Harmon et al. (2025) which include phase equilibrium approaches were added, and show uncertainties too high to resolve whether the magma is at lithostatic pressure.

We added to line 54: "Pressure estimates from projection on the haplogranitic ternary suggest < 50 -100 MPa, while Rhyolite-MELTS yields 44 ± 11 MPa to 47 ± 32 MPa depending on oxidation state, all of which require an assumption of volatile content, and fail to model this system in that they produce quartz + plagioclase \pm orthopyroxene, when the observed phases are plagioclase + two clinopyroxene¹⁸."

5. For understandable reasons, the focus of the paper is more on pressure than on temperature, because pressure is admittedly a more difficult parameter to assess. Yet, the reported spread in pre-eruptive temperature is quite large (70°C). If at equilibrium, such a spread should come along with varying crystallinity (for constant bulk rock composition) which I understand is not observed (true?). Alternatively, besides model errors, this spread may simply reflect magma mixing, as is common in many volcanic areas. There is nothing wrong with this but in my opinion it introduces an additional source of uncertainty, given the quite strong temperature dependence of H₂O solubility in silicic melts. I am not sure (but I may have missed this point) that the modelling performed takes this explicitly into account?

Perhaps this was unclear in the text. The 70 °C window is an uncertainty on the temperature calculation on the basis of petrological constraints, not representative of a spread across the samples. Also, the crystals represent such a small volume fraction of the magma that many of the small glass chips are aphyric, so it would not be possible to search for systematic differences in crystallinity across chips. The chips don't show large variability in major or trace elements that would suggest unequilibrated magma mixing, although some crystal transfer between the melt and an overlying, low-degree partial melt, felsite has been proposed. We chose a smaller range of temperature to present in the paper (900-920 °C) that was consistent with experiments conducted on this composition which become too crystal rich at lower temperature (e.g., Masotta et al., 2018; and require higher temperatures at lower pressures, meaning drier initial

conditions). But, the sensitivity to temperature in this range is much lower than the sensitivity to pressure, so for 45 MPa at 900 °C (the result we present) the starting volatile budget at saturation is 1.77 wt% H₂O and 108 ppm CO₂, and at 850 °C for the same water content (which can't vary much and still match the observed vesicularity and remaining water in the glass), CO₂ only goes up to 121 ppm. Not enough to make a difference in the conclusions. We expect the temperature has a small effect on the results, below our uncertainty to matching the measurements, and therefore we don't try to place additional constraints on the starting temperature – this is the goal of separate, ongoing experimental investigations in our groups and the preliminary experimental results, where glass chip stability is tested at different P-T-X conditions (da Silva et al, in prep) support a narrow range of possible storage temperatures.

The background on existing work on the thermometers has been expanded which clarifies the reason for the large range on line 51-54: “various geothermometers yield different estimates and uncertainties, from two-pyroxene equilibration between 920-940 °C^[1,16] and 890-910 °C^[12,17], while the phase assemblage places the weak constraint between 800 and 950 °C^[1].”

We add the limitation of temperature to the text on lines 219-222: “Starting from a cooler (~850 °C) initial temperature would have only a modest effect on starting CO₂ (121 vs. 108 ppm at 45 MPa and 1.77 wt% H₂O), and so this method cannot place tighter constraints on the starting temperature compared to existing geothermometers and experimental studies³².”

6. In the abstract, it is stated that disequilibrium crystallisation during ascent complicates the unravelling of storage conditions. Again here, a word of caution is in need: while this is true for slowly ascending silicic melts, in those ascending fast, ie Plinian eruptions, very little happens from the viewpoint of crystallisation, and the preserved crystals (phenocrysts along with their melt inclusions) can be used to restore storage conditions, or any pre-eruptive phenomenon (mixing etc..). So this statement is incorrect or belongs to a specific class of eruptions, which should be termed to avoid confusion or misunderstanding from untrained readers.

This is a broad summary that includes both crystallization and vesiculation which matters for inverting volatile concentrations in even fast-ascending eruptions. We also point the referee to recent studies that suggest nanolite crystallisation may be important even in explosive eruptions (during rapid ascent), which do impact melt properties (e.g., Caceres et al., 20204: <https://doi.org/10.1038/s41467-024-44850-x>) and argue for caution in interpreting storage conditions.

7. It is unclear to me how CO₂ diffusion is handled in the model of H₂O diffusion described in the SI. The authors state that CO₂ diffuses slowly compared to H₂O, which is true for H₂O-rich conditions but no more at low water contents (< 1 wt% H₂O) where both species have similar diffusivities, CO₂ diffusing even faster than H₂O at very low H₂O content (which equation do they use? I think it is from the compilation of Zhang et al when looking at their codes but it would be better to mention that explicitly in the SI as well). It may be that this will have a marginal effect of model outcomes, but I suggest this point be better explained and perhaps elaborated upon.

We never reach low water contents (i.e., where CO₂ diffusivity matches or exceeds that of H₂O). The lowest water contents we consider are still above 1.5 wt% H₂O. We indeed use the model of Zhang et al. (2007), we apologize if this was unclear. The reference has been added to the methods.

The reference was added to line 329.

Referee #2:

1. Significance

This study makes a significant contribution by leveraging direct sampling from a borehole, an approach uniquely suited to constraining sub-volcanic systems. As the authors rightly note, such systems often remain poorly resolved by geophysical methods. The use of physical samples enables insights into the magmatic plumbing system that would otherwise remain speculative. Extracting meaningful signals from these samples requires sophisticated modeling approaches to disentangle the effects of natural processes from those introduced during acquisition and recovery. The authors' effort to meet this challenge is hampered by lack of support for a key assumption: fluid saturation of the magma at the storage/recovery depth.

We apologize if this was in some way unclear. The purpose of this paper is to investigate the saturation pressure; We don't assume fluid saturation, but rather it is the result of the paper. When simulating decompression from 35 and 45 MPa (below lithostatic pressure) we are directly looking at the evolution of the melt from a point of undersaturation with respect to lithostatic conditions. We describe the results in terms of the equilibrium pressure to which that volatile load corresponds for convenience. Using this method, we cannot distinguish between an undersaturated magma and saturated magma that is being stored under-pressured – given that vesiculation begins only when the system pressure drops below the saturation pressure, but we find that neither volatile-poor scenario matches the observations. The “assumption” of equilibrium with a particular pressure in theory places an additional constraint on the H₂O-CO₂ ratio in the melt and vapor phases, but because we find that H₂O is tightly constrained by the final melt and vary the vapor phase composition (H₂O-CO₂ ratio in the melt) to match the final water, we are really only varying the CO₂ content which would be the free variable without this assumption.

We have modified the following lines to make this clearer throughout the text:

Lines 73-75: “These measurements have been interpreted to suggest that the magma is either 1) stored and degassed to equilibrium at a pressure less than lithostatic^[1,12] or 2) originally undersaturated^[14,16].”

Lines 151-160: “Because we do not simulate the exact fragmentation dynamics, we consider a range of possible decompression rates from the initial volatile saturation storage pressure (35-55 MPa), evolving towards hydrostatic (16 MPa) over 10 s, 30 s, 1 min, 5 min, 30 min, and 1 hour, corresponding to decompression rates 5.3-2900 kPa/s. The initial conditions are specified by choosing a starting volatile saturation pressure and molar ratio of water and CO₂ in the co-existing vapor phase, which together set the total water and CO₂ volatile contents. With a vesiculation model we cannot distinguish between the two competing hypotheses of an undersaturated magma (which produces a first volatile phase when the system pressure drops to the saturation pressure), and magma stored saturated, but at less-than-lithostatic pressure with a negligibly small proportion of co-existing vapor.”

Line 188: specify “initial saturation pressure”.

Line 215: specify “storage saturation pressure”.

Line 263: specify “volatile-saturated” and lithostatic.

2. Validity of Methods, Quality of Data, and Quality of Presentation

The authors employ a suite of plausible and technically sophisticated models, including H–O speciation, $\text{H}_2\text{O}+\text{CO}_2$ fluid solubility, and viscous flow dynamics, to interpret the volatile content of the recovered glass.

Minor weaknesses: (a) the application of two-clinopyroxene thermobarometry is too imprecise to resolve pressure differences over the small depth range in question. More broadly, any barometric approach relying on clinopyroxene is likely to be similarly limited in precision under these conditions. (b) Regarding the data presentation, the BSE images confirm the presence of fresh glass and two mineral phases, but they do not confirm the stability of the mineral assemblages invoked in the thermobarometric models.

We give the thermobarometric study for context in the existing literature. We agree that the constraint it imposes has a large uncertainty, and the primary purpose of our study is to investigate volatile saturation and the pressure conditions of the chamber through the numerical model for comparison to the known depth. See the response to Referee 1, point 1 above.

The following text was added to line 50-58: “Despite the direct sampling of the glass, established methods to determine pressure and temperature have yielded poor constraints: various geothermometers yield different estimates and uncertainties from two-pyroxene equilibration between 920-940 °C^{1,16} and 890-910 °C^{12,17}, while the phase assemblage places the weak constraint between 800 and 950 °C¹. Pressure estimates from projection on the haplogranitic ternary suggest <50-100 MPa, while Rhyolite-MELTS yields 44±11 MPa to 47±32 MPa depending on oxidation state, all of which require an assumption of volatile content, and fail to model this system in that they produce quartz + plagioclase ± orthopyroxene, when the observed phases are plagioclase + two clinopyroxenes¹⁸.”

Given that the mineral-mineral thermometers are not the focus of this study, but part of the background literature and provide only a rough starting point for the inversions, which are better constrained by the existing experimental work done directly on Krafla samples, demonstrating the co-existing pyroxenes is beyond the scope of the study.

3. Appropriate Treatment of Uncertainties

The manuscript would benefit from acknowledging that volatile saturation calculations provide only minimum melt storage pressure. The calculated volatile saturation pressure of 35 MPa is a lower bound; if the magma were not fluid saturated, then the actual storage pressure could have been significantly higher. Critically, the manuscript does not present compelling evidence that the magma was indeed fluid saturated at the storage depth, which is a foundational assumption of the decompression modeling. Demonstrating fluid saturation typically requires independent evidence, such as the presence of melt inclusions containing sufficient volatile concentrations to require saturation, or intact fluid bubbles within inclusions that have not been altered or decrepitated.

Our results are compatible only with a scenario in which the magma is both saturated and stored at lithostatic pressure as either case of 1) volatile undersaturation for a magma stored at lithostatic pressure or 2) a magma stored under-pressured are both inconsistent with the final measured glass chemistries after correcting for the response of magma to drilling. We haven’t assumed this, but rather directly asked whether either of these scenarios can explain the data.

There has not been a sufficient, reliable population of melt inclusions isolated to investigate volatile contents from earlier in the magma history, but we would argue the presence of the high bubble number density found in the recovered chips indicates that the magma cannot have been far from saturation or syn-decompression cooling would have prevented vesiculation. See response to Referee 2, point 1 above.

4. Validity of Conclusions

If the magma was fluid saturated at the lithostatic pressure (e.g., ~52 MPa), the presence of lower volatile concentrations (corresponding to 35 MPa saturation pressure) in the glass would necessitate thoughtful modeling of degassing and resorption processes, as pursued in this exercise. However, in the absence of firm evidence for fluid saturation at the outset, the necessity of such modeling disappears. That is, if the magma was undersaturated at the time of recovery, there would be no need for dynamic pressure 'buffering' during ascent to cause reabsorption of volatiles. This alternate assumption, volatile undersaturation, emerges as a plausible and perhaps even more consistent explanation for the observations than the authors' original premise of fluid saturation. This is itself a noteworthy result, given the challenges associated with critically determining fluid saturation state in natural systems. If the authors reframe their findings accordingly, the impact of the study is still compelling.

In addition to above, we apologize if the discussion around resorption was unclear. We invoke resorption that is triggered by the change in solubility of the melt in response to cooling, not the pressure changes; the new figure may help clarify this point. The observation of a relatively homogeneous population of glass chips with drilling time over several hours and bubble number densities indicative of rapid decompression rates requires that the magma pressure be maintained away from the drill string. A volatile undersaturated magma would have insufficient CO₂ to match the joint constraints of vesicularity, and water and CO₂ contents of the final glass.

See the above responses to Referee 1, point 1 and Referee 2, point 1 regarding undersaturated starting conditions.

I acknowledge that I have not reviewed all of the references cited in the manuscript about these drill cores, and it is possible that my concerns are allayed therein. Nonetheless, I hope the comments provided here will be helpful for at least highlighting areas of potential confusion that other readers may also encounter.

5. Suggested Improvements

Several aspects of the manuscript would benefit from clarification and revision, regardless of the fluid saturation assumption. The attached PDF includes many comments and suggestions for improvement, keyed to lines of text and figures.

We thank the reviewer for the detailed comments on the text.

Line 12: Suggested delete "arguably".

Accepted.

Line 16: Suggested delete "careful".

Accepted.

Line 18: “previous assertions based on classic methods¹”. This is vague. Instead, say what the previous assertions were. Presumable non-lithostatic, so “pressurized”? I’m not sure what this means.

Amended to read: “unlike previous assertions of lower vapor pressures based on classic methods.”

Line 46: “Various geothermometers vary”. Awkward.

Amended to read: “Various geothermometers yield different estimates and uncertainties.”

Line 48: “corresponding pressure”. What is the evidence that the magma is saturated at the storage area with an H₂O-Co₂ fluid? Typical evidence includes: melt inclusions with sufficient H₂O+CO₂ to require saturation at the lithostatic pressure, the presence of fluid bubbles inside melt inclusions that have not undergone decrepitation or alteration, etc.

See above response to Referee 2, point 1: Significance.

Line 48: “calculated from volatile saturation of the melt”. Be more specific about this pressure modeling based on volatile saturation. What H₂O and CO₂ and temperature combinations yield this range of pressures?

This comment seems to be asking about the compiled data from Zierenberg et al. (2013) and Watson (2018) which we can have been added to the Extended data. We also added a table of the model input conditions for clarity.

Line 49: “significantly below the lithostatic pressure (~50-57)”. The volatile saturation pressure yields only a minimum for the storage pressure. If the magma is not volatile saturated, the magma could be stored at a pressure above this value.

See response to Referee 2, point 1 above regarding the two competing hypotheses. We hope these are now clearer in the text.

Line 50: “hydrostatic pressure of the well (~16 MPa)¹”. Is the rock at 2000 m both porous and water saturated?

We give the hydrostatic pressure because that is certainly the pressure in the well. Generally, our results are consistent with at least the rock directly overlying the magma (felsite with small degree of partial melt) to be impermeable (lines 228-231).

Figure 1: Figure 1 shows the lithostatic pressure at the recovery depth is ~52 MPa.

That is correct. In the text we report a range (50-57 MPa) due to uncertainty on the depth, but we can only represent one value for density to scale between depth and pressure in the figure.

Two cpx modeling is not mentioned in the text and would anyway be far too imprecise over this depth range for resolving pressure in a meaningful way. Any mineral barometer assemblage involving cpx is going to be similarly futile.

Agreed, these are literature values now discussed with greater context (see response to Referee 1, point 1 above). We plot the temperature estimates of the magma on the depth y-axis in red because they were recovered from that depth. The pressure estimates in blue all correspond to the secondary y-axis and show pressures up to ~30-50 MPa.

This has been clarified in the figure caption which now reads: “Temperature (red) with depth in the borehole⁶, vs geothermometric constraints from augite-pigeonite, and clinopyroxene-orthopyroxene-plagioclase-magnetite-illmenite, Rhyolite-MELTS¹ plotted at the recovery depth, with a secondary axis showing lithostatic pressure at the corresponding depth, and saturation pressure (blue, from VolatileCalc^{1,16}).”

These BSE images confirm the presence of nice clear glass and presence of 2 mineral phases, but they don't confirm the presence (and stability) of the mineral assemblages called in the thermobarometers. It's hard to tell whether the voids in panel c are vesicles or just pores introduced during preparation of the surface. The glass chip in panel e doesn't seem to have any vesicles. The emphasis on bubbles at this point in the text is hard to reconcile with the image data.

There are some fractures in panel d, but we interpret the small voids to be vesicles (~12%) and they are consistent across multiple chips and collaborators. We would argue that panel e shows a small fraction of vesicles (~0.5%). We show here the full range of vesicle textures that are present in the chips, but emphasize that throughout the manuscript we are discussing a typical vesicularity of only 5%. We do not intend for the chips presented here to argue regarding whether the mineral-mineral pairs are in equilibrium, which in any case do not give a tight-enough constraint to answer our questions about storage conditions.

Clarified the figure caption (line 92-93) to read: “IDDP-1 glass chips that show the variability in vesicularity.” We added labels in the figure of our interpretation of the texture.

Line 54: “palgioclase”. Typo.

Corrected.

Line 57: Excellent point, effectively made.

Thank you. We are glad this was clear to the referee.

Line 64: “bubble growth model”. The characteristics of bubbles are not determined herein, although later (l. 68-75) they're described as having been quantified. After reading this section, I'm wondering about the utility of the bubble growth modeling in the context of the fragmentation mechanism.

The bubble quantification was not performed here, but cited from existing literature to show the constraints we can place. We directly use the bubble number density in the model, and consequently (by matching number and total vesicularity) the model reproduces the bubble sizes. From the referees other comments, it seems that the fragmentation mechanism of thermal granulation was not clear and the referee is assuming fragmentation via bubble connectivity. We have left the specific mechanism vague here until it becomes relevant to the cooling below. We amended the preceding paragraph (line 82) to read “against the measured glass chips” to provide further a smoother transition that justifies the need to report these values. And clarify on line 116 that for the purposes of thermal modeling the fragmentation could occur “by any mechanism” which is specified to be thermal shock granulation on lines 144-146.

Line 73: “OH/H2Om”. Should be possible to infer temperature from the speciation for known total H2O?

Speciation is highly sensitive to the cooling path and happens rapidly enough to keep pace with the cooling unless 1) the cooling is extremely rapid (only achievable in laboratory conditions) or

2) the diffusion and reaction rate become very slow as the melt viscosity increases which in our case happens about 100 °C above the glass transition temperature. We show results for this.

Line 78: “measurement of the natural glass”. I’m not sure which measurements listed above constrain the cooling rate of the glass; are the OH/H₂O_m speciation data being used, as I suggested above?

This constraint comes from geospeedometry results from Wadsworth et al. (2024), described two sentences later, which has no explicit modeling in it. Differential scanning calorimetry was performed that allows for determination of the glass transition temperature which depends on both melt chemistry and cooling rate. An initial measurement is performed on the natural glass and then the sample is repeatedly heated and cooled across the glass transition at different rates. The glass transition temperature of the initial sample can then be compared to known cooling/heating rates which reveals the cooling rate of the sample. Clarified to read: “Geospeedometry via differential scanning calorimetry indicates...”

Line 81: “Thermal analyses”. Here, clarity is needed to resolve modeling from data. Clearly, geospeedometry is a modeling effort, but what are “thermal analyses”?

These are the DSC measurements described above.

Line 83: “cooling rates far exceed those measured in the glass”. Does this refer to the measured temperature in the borehole? How was cooling rate ‘measured in the glass’?

These are the cooling rates measured for the glass itself from the DSC. The water in the borehole remains cold at ~25 °C throughout drilling (see Mortensen et al., 2010 for more information on the temperature above the magma).

Line 87: “fracturing or fragmentation of the material”. Seems like the presence of fragments would be corroborated by the state of the material recovered (shard shapes), unless this is obscured by the drilling process.... clarify.

There are some large glass shards recovered, and some which show evidence of interaction with water over the whole surface, but any quantitative estimate of the size distribution of the fragments would indeed be overprinted by secondary drill action.

Line 89: “magma fragments during the onset or early progression of cooling”. As above, if fragmentation occurred naturally, wouldn’t bubbles exist or shard shapes testify?

See above.

Line 92: “perlitic glass^{24,25}”. This is the inverse of what you’d expect from fragmentation caused by impingement of bubbles. I question the relevance of perlitic glass to this geological scenario.

We apologize if this section was confusing or suggested that we envision fragmentation to occur by bubble impingement. Instead, we are advocating here for a thermal granulation process which is also the process thought to produce perlitic fracturing. See clarification above.

Line 97: “volatile composition and vesicularity of the glass changed only modestly”. What is the modest change? Any evolution in the volatile composition and vesicularity of materials recovered during drilling is new information, not presented above.

We refer to Saubin et al. (2021): the populations overlap, but the proportion of shards with vesicularity >3% actually decreases with drilling time.

Line 98-100: “the decompression could be offset at increasing depth by pressure drop generated by viscous resistance to flow up the borehole”. After reading the entire ms, I return to question the need for this modeling. Unless it can be demonstrated that the magma was vapor saturated at the recovery depth, then there is no need to buffer the pressure and drive reabsorption of volatiles during ascent through the borehole.

See response to Referee 2, point 1 above regarding undersaturation.

Line 107: “35-55 MPa”. The range of lithostatic pressure based on recovery depth and crustal density is 50-57 MPa, with 52 MPa shown in figure 1. Why is 35 now invoked as a relevant “storage” pressure?

We investigate the range of saturation pressures measured directly from the glass. See clarification detailed above in response to Referee 2, point 1.

Line 110: “most magma may continue to be stored at pressure until the onset of cooling”. This phrasing is awkward because of the word “stored” which suggests it’s not moving, when I believe the proposed mechanism of pressure buffering is the result of magma flowage.

Rephrased to “most magma may continue to experience near-storage pressures until the onset of cooling.”

Line 116: “non-explosive fragmentation from thermal shocking”. Fracturing caused by introduction of drilling fluid? That is plausible.

We are glad the reviewer finds this hypothesis plausible.

Line 124: “hydrostatic”. Not sure of the relevance here of hydrostatic pressure, given that there’s been no evidence presented of groundwater-fluid saturation in the region above magma recovery.

The pressure in the borehole is hydrostatic. This is stated on line 72-73.

Line 126: “initial conditions are water saturated for a specific molar ratio of water and CO₂ in the co-existing vapor phase”. Have I missed the evidence for vapor saturation that would justify this assumption?

See above response to Referee 2, point 1.

Line 129: “45”. Seems like a value was selected falling halfway between 35 and 55, but still not sure why 35 MPa is relevant. After reading all, it seems like 35 is a foil, included to demonstrate what model failure looks like.

35-45 MPa is the range of volatile saturation pressure reported in previous works. See the changes detailed above regarding the starting conditions.

Line 160: “constrain magma storage to lithostatic pressure”. This figure is confusing. The pairings of P and T are given in the text, but there is no mention of temperature in the figure or caption. What is the significance of the phrase “constrain magma storage to lithostatic pressure”? Are the models supposed to help decide whether the initial pressure was 35, 45, or 55 MPa? If so, I suggest an alt caption title: modeling of evolving speciation of H-O and CO₂ abundances to determine timescales of decompression for three starting pressures.

We hope in the figure to draw the reader to the starting saturation pressure (which is what we can constrain), we have added the remaining model input parameters to Extended Data Table 1. We also want to emphasize that the joint constraints of all the chemistry and texture together is a highlight of our model that places better constraints on both initial conditions, and decompression and cooling rates. Rephrased to “Simulations of vesicularity need to be combined with water and CO₂ to distinguish different P-T paths, which result in initial volatile-saturated, lithostatic conditions.”

Line 193-196: “The magma cannot have undergone slow decompression for tens of minutes to hours during approach of the drill string, suggesting the roof rock is capable of isolating the magma from the surrounding hydrothermal system, at least on short timescales.” Agreed, this claim is well supported.

We are glad the referee was convinced by this argument.

Figure 4: This is plausible.

We are glad the referee was convinced by this argument.

Line 322: “Code Availability”. I encountered a few difficulties executing the reviewer copy of the code, which I documented in screenshots. 1. Some things are a little unclear (how to specify location of results, as directed in the instructions when user clicks “run”). 2. The only output I saw was the set of “t = ” values at each iteration written to the window; output did not include plots, and instead there were error codes. 3. Message in window “no preview available (the file is too large) gave the options of downloading a file or opening in a new window. Both options prompted me to download a .mat file, which I can open in Matlab (of course), but is not illuminating with respect to output.

We apologize that the instructions for running the code were not clear. The results are directly written to the “results” folder as required by the Code Ocean repository, in which the data file header contains information about the run conditions which are also stored in the .mat file. The results are fairly large and so are only saved to the file, not output to the command line where only a progress report is shown stating the simulation time that has been reached. The warning codes can be safely ignored. We have added a readme to the repository along with the visualization scripts.

Extended data figure 1:

1. Use units consistently. here using minutes in caption and seconds in the legend. indicate the important distances (0.03 and 0.13) with lead-line labels in the figure panel. Also, best practice is to use SI units not cgs units.

Thank you for the attention to detail in the figure. We have changed everything to cm for consistency. The purpose of having cooling rate in min is to match the DSC measurements which are reported in K/min and keep the values easy to read and interpret.

2. The legend implies a continuous spectrum of times (0-5 minutes/ 300 s), yet the curves are discrete values of time and none of the curves approaches the upper end of the spectrum, judging by the colors. Why not just use a simple legend that defines the modeled time scale values? Doing so would increase clarity of the figure and match the caption.

Time is calculated continuously in the simulation and we show a subset of the times, we prefer to keep the continuous color bar to allude to the continuous nature of the field. Thank you for

noticing that the full range of the color bar was not being used, the missing curves have been added so they now span the full time range.

3. why no curve representing on the cooling rate panel to match the one on the temperature panel; assume it would be a similar horizontal line at 0.

Yes, it's instantaneously very high (limited by the discretization in time) and then uniformly 0 so, the limit is not really physical.

4. Clarity of message: it's hard to see that any of the stars is plotted at a distance of 0.03 cm as suggested by the caption as a viable solution. Is that where the lightest orange curve intersects the bottom of the gray horizontal bar? if so, leader lines would help.

The missing shaded bar of matching cooling rates have been added.

Extended data figure 2:

1. The legend is in reverse value order from previous figure: increasing downward. Suggest keeping the orientations consistent.

Thank you, we have implemented this suggestion.

2. Similar comment to Ext F1, regarding use of a spectrum of colors for the legend that is not invoked in the plotbox, and because of this needing to articulate the values explicitly anyway in the caption.

Same as above.

3. Here, the distances are given in mm. Suggest matching the units across these figures (preferring mm over cm) for clarity and coherence.

We have chosen to keep the annotations in this figure in mm to small decimals.

Line 513: "we require". Use of first person "we require" is odd here, given the ease of using third person in this situation.

Accepted.

6. Clarity

The manuscript's clarity would improve with more consistent treatment of key variables. The selection and use of pressure values are sometimes ambiguous—at times treated as constraints derived from observations (e.g., lines 48–49), and at other times as model variables or assumptions. A similar issue arises with temperature, which is alternately described as a measured parameter (e.g., lines 78, 81) and a modeled outcome (e.g., line 79). These inconsistencies make it difficult for the reader to follow the logic of the analysis.

We use the pre-existing constraints to select an appropriate range of pressures to test with the numerical simulations, we hope this is clarified now in the text. See response to Referee 1, points 1 and 5 above regarding the discussion of the pressure and temperature constraints from the field:

The following text was added to line 54-58: "Pressure estimates from projection on the haplogranitic ternary suggest <50-100 MPa, while Rhyolite-MELTS yields 44±11 MPa to 47±32

MPa depending on oxidation state, all of which require an assumption of volatile content, and fail to model this system in that they produce quartz + plagioclase ± orthopyroxene, when the observed phases are plagioclase + two clinopyroxene¹⁸.”

And to line 51-54: “Various geothermometers yield different estimates and uncertainties, from two-pyroxene equilibration between 920-940 °C^[1,16] and 890-910 °C^[12,17], while the phase assemblage places the weak constraint between 800 and 950 °C^[1]”

In lines 104-109 we rearranged to place the measurements from the natural glasses (via DSC) first and then describe the modeling: “Although the cooling from the drilling fluid should be rapid, we use a thermal model to seek paths consistent with measurements of the natural glass; geospeedometry via differential scanning calorimetry indicates that the IDDP-1 magma cooled through the glass transition regime (T_g) at 7-80 °C/min with a T_g of ~480 °C^[27]. We begin with one-dimensional cooling in a planar geometry from an initial 900 °C via conduction and forced convection at the surface with steam (400 °C and 16 MPa) where drilling fluid is in direct contact with the melt.”

Referee #2 (Remarks on code availability):

I encountered a few difficulties executing the reviewer copy of the code, which I documented in screenshots.

We apologize if the instructions for running the code were unclear. See below for specific responses.

1. Some things are a little unclear (how to specify location of results, as directed in the instructions when user clicks "run").

The Code Ocean service requires the results to be written to “results”, which the code is set up to do and then saves under a title string that includes the run parameters. We added a readme with further documentation. Otherwise, the messages that appear asking about the results location can be ignored, it will save to that directory.

2. The only output I saw was the set of "t = " values at each iteration written to the window; output did not include plots, and instead there were error codes.

This is correct, the only outputs to the command line are a progress update on the simulation time and there are some warnings that come only from the plotting and can be safely ignored. We suppressed them in the code. It looks to have run correctly in the attached screenshots.

3. Message in window "no preview available (the file is too large) gave the options of downloading a file or opening in a new window. Both options prompted me to download a .mat file, which I can open in Matlab (of course), but is not illuminating with respect to output.

That file contains the simulation results. We added the code with the simulation visualization to the code repository.

Response to editor and reviewers

Editor:

Length: At about 2900 words, the present length of your main text is somewhat longer than we normally allow for a short *Nature* Article with five small display items (that is, figures and tables). We do, however, have some flexibility and could allow you another 100 words or so to address the referees' comments, but we will need to consider this a firm upper limit.

We have added a net 79 words to the main text and 10 words to the figure captions.

Title: The present title of your manuscript is somewhat longer than we can accommodate. Would simply "Disequilibrium response to tapping crustal magma reveals storage conditions" suffice? Please feel free to suggest an alternative title, bearing in mind that it should be 75 characters or less in length (including spaces) and not contain punctuation.

We are amenable to this change, it has been updated in the manuscript.

Order: The order in which your text file (without embedded figures) should be presented is as follows: title, authors, authors' affiliations (with full address information), main text, references from main text, main figure legends, online Methods section, Data and Code Availability statements, references only cited in the online Methods section, acknowledgments and author contribution statements, and then Extended Data Figure/Table titles and legends.

Please supply the five main figures as separate files in editable format (**not** flattened) and each of the Extended Data Figures/Table as separate files in JPEG, TIFF or EPS format (all without titles or legends). Please also ensure that you cite each Extended Data Figure/Table, in order, somewhere in the main text or Methods section (as "Extended Data Fig. 1", "Extended Data Table 1", etc.).

We have added the full address information to the author affiliations and re-ordered the sections as outlined above.

Sub-headings: In our new all-Article format, we encourage several sub-headings within the main text of short Articles to break up the text. Such sub-headings may only be up to 40 characters in length, including spaces.

References: Please move the date of publication to the end of each reference. Also, please ensure that all references contain full publication information (particularly for the PhD theses cited).

Missing DOI's and ISSNs (when no DOI was available) have been added, as well a link to the PhD thesis. The multi-authored "thesis" is in fact a conference abstract presenting the contents of the thesis which has been corrected.

Figure legends: Please define the error bars in each figure within its legend (for example, do they represent 1 or 2 standard errors, or 95% confidence limits, etc.?).

They are 1σ errors reported in the original papers (Zierenberg et al., 2013; Watson, 2018). This has been added to the figure captions. In doing so, we caught a minor numbering issue in the reference list introduced in the previous round of revisions when new references were added. This has been corrected in Fig. 1 and Fig. 4 captions. All other references were manually double-checked and are correct.

Extended Data: Nature is now integrating the supplemental figures and tables into the final version of most papers. Extended Data do not appear in the printed version of the paper but are included

online within the full-text HTML and at the end of the online PDF. Extended Data are an integral part of the paper and only data that directly contribute to the main message should be presented. All Extended Data must be referred to in the main text, figure legends and/or Methods section, and their figure legends should be listed sequentially at the end of the main text, not in the Extended Data files. Authors should assemble the Extended Data into a maximum of ten, A4 size, multi-panelled display items, submitted as individual JPEG, TIFF or EPS files. They must be provided at the same quality as figures for print, but there are important differences in their formatting. More specific instructions are provided in the Extended Data Formatting Guide (<https://tinyurl.com/y8ktz4z6>).

We have changed the panel labeling to lower case letters for all the extended data figures and updated the formatting on the extended data table.

Source Data: To further increase transparency, we encourage authors to provide, in spreadsheet form, the data underlying the graphical representations used in the figures. This is in addition to our well-established data-deposition policy for specific types of experiments and large datasets. Readers of the online manuscript will be able to access the Source Data directly from the figure legend. Spreadsheets can only be submitted in .xls, .xlsx or .csv formats. One file per figure is permitted; thus, if there is a multi-panelled figure the Source Data for each panel should be clearly labeled in the csv/Excel file; alternatively the data for a figure can be included in multiple, clearly labeled sheets within an Excel file. File sizes of up to 30 MB are permitted; however, it is expected that the vast majority of Source Data files will be considerably smaller than this. When submitting these files with your manuscript, please select the file type “Source Data” and use the title field in the file description tab to indicate the figure to which the Source Data pertain.

We have provided spreadsheets for the data in Fig 2-4. Fig.1 contains no original data, and is better found in the original cited publications, and Fig. 5 is a schematic. A portion of the data for Fig. 3 was down-sampled to one-half resolution to stay within the file size limit (the time series are much higher resolution than can be seen on the figure), but the full-resolution data is available through the Code Ocean repository.

Referee #1:

I have carefully read the revised version of this paper, and the accompanying rebuttal letter, and found that the authors have correctly answered the questions raised during the first round of review, clarifying the most important issues I could see. I thank them for their efforts in explaining some aspects in a very detailed way. I am happy to recommend acceptance of this revised version.

I have made a few minor additional comments/remarks reported below, and keyed to the line number of the manuscript.

We thank the referee for their time and attention to this manuscript.

52-53. Note that the « weak » constraints on temperature are very specific to this Icelandic system, but cannot be taken as representative of T estimates elsewhere. Again quoting Mt St Helens or Pinatubo, which represent a kind of canonical studies, temperature constraints gained from petrology are quite accurate

We refer here specifically to the measurements on the Krafla system. While calibration data for geothermometers have reported high precision (<10 °C) in these well-studied systems, they can be challenging to apply to systems with different chemistry or when an independent temperature estimate is not available for validation. Additionally, the natural variability between individual mineral grains is often larger than this precision.

57-58. That is correct.

We are happy the referee agrees.

62-70. A word of caution/comment here. It maybe that melt inclusions are inappropriate, or cannot be found in abundance, at Iceland, but they were quite appropriate at many other volcanic centers to set tight constraints on pre-eruptive storage conditions, in particular when combined with phase equilibrium considerations. Saying that all melt inclusions result from disequilibrium is an extreme statement: some of them may indeed grow during decompression, but some other just grow in the reservoir and are likely to record storage conditions (for instance the melt inclusion with the highest CO₂ content in figure 4). I also point out that in several instances melt inclusions do occur along with fluid inclusions, and this is taken as evidence of fluid saturation, in which case the H₂O and CO₂ systematics gained from melt inclusions analyses correctly inform about P (and not just on minimum of P storage. As written now, the text may give (to readers unfamiliar with this theme) the wrong impression that crystals in magmas everywhere cannot be trusted, and that only the matrix glass is likely to inform about storage. I would slightly reword this section by making it clear that this (restriction) is specific to the case study.

1) We can infer that the rate of entrapment is likely higher during rapid (disequilibrium) crystal growth and we should therefore treat melt saturation pressure population distributions with care – a large concentration at a particular pressure/depth does not necessarily imply long-term storage at that location, only a high proportion of the crystal growth.

2) While melt inclusions do sample the system at different locations, which likely includes some in storage, we cannot *a priori* determine whether a melt inclusion was formed during transport vs. storage. I.e. even if a melt inclusion saturation pressure matches the depth of a known storage region, we cannot distinguish between a melt inclusion grown under fluid-saturated conditions during storage vs. one entrapped deeper in the system where the fluid was undersaturated.

3) As you say, melt inclusions which co-occur with large fluid inclusions are an indicator of a co-existing vapor phase, but that does not imply equilibrium between melt and fluid, nor does it preclude continued interaction between the co-existing fluid and melt (and also the crystal host) during ascent and cooling. Given the unknown initial pressure/size of the vapor bubble, even co-entrapped phases are challenging to place accurately to a depth/pressure within the system.

Part of what we hope the reader takes away from this paper is the importance of considering the full path of a sample and the measurements we make on it before we interpret what it tells us about the system; this applies to melt inclusions as well as matrix glasses. We expect that some readers may disagree with these claims about melt inclusions, which are intended to give context about the need for and novelty of this study, and represent the authors' interpretations. This is, admittedly, not the place for an extended discussion on the validity and applicability of melt inclusion data and interpretations, which we will leave for future work.

Additionally, we would like to clarify that we don't present any melt inclusion data in our manuscript, the measurements in Fig. 4 show the variability left in the matrix glasses.

We have amended the text on line 69-72 to read "are produced preferentially during disequilibrium crystal growth" and "and therefore are poorly suited to ~~cannot~~ investigate equilibrium storage."

67-68. Assuming that some melt inclusions were trapped during ascent is equivalent to say that some crystals grew during ascent. But are the time scales of magma ascent and crystal growth comparable?

Broadly, this will depend on the system. When we are discussing transport/ascent in the context of understanding storage in the shallow crust, we mean transport in the deeper part of the system (mantle to the shallow crust) rather than ascent to the surface during an eruption (which the IDDP-1 chips have not experienced). In the deeper part of the system ascent rates are quite unknown and temperatures are high, and the process may involve multiple stages of stalling, so it's plausible that crystals are growing during ascent. In the specific case of IDDP-1, there is no substantial crystal growth during quenching from the drilling fluid and the crystals could have grown during transport or storage; they preserve zonation indicating previous growth at higher temperatures. Where that growth has occurred is outside the scope of this study.

122. Something important to take into consideration here is that the magma will flow upward using the fracture path made by the drilling itself. In nature, magma has to break-up frontal rocks itself and this will consume energy as well as slow the process.

We very much agree. The coarse estimate here of decompression along the pipe is for the magma flowing in the tightly confined borehole – the diameter is only 0.15 m, much narrower than natural conduits (>>5 m). This section is only intended to discuss the IDDP-1 chips, not natural systems.

130-131 ; A sensitivity analysis on the role of melt viscosity would be useful here.

The pressure drop is directly proportional to melt viscosity, so it has a linear relationship. This is a scaling-type analysis so a factor of 2 in viscosity, for example, would change this length scale also by a factor of 2. We added to the text on line 126-127: "This lengthscale would become shorter directly proportional to an increase in viscosity."

141-142. Please give a reference for this threshold of shear driven fragmentation. Perhaps Spieler et al. 2004?

This is based on the criteria in Webb and Dingwell (1990); we added this reference on line 135.

241-243. This is an important conclusion in particular for felsic systems. The high viscosity of rhyolites is often taken as evidence for their enhanced potential (relative to mafic melts) to preserve HT processes which is clearly not the case.

We are glad the referee sees the importance of this claim.

259. ...during rapid cooling and quenching at depth.

Added, now on line 231-232.

440 . This is a PhD thesis. Is it available to a large audience?

This is a PhD thesis and is publicly available through the university database. A doi has been added to the reference, now on line 305-307.

448. This also looks like a PhD reference but is multi authored? The same remark applies as to the availability of that work.

Yes, sorry for the confusion on this reference. There is a PhD thesis for this work, but the reference actually should point to the publicly available EGU abstract. This has been corrected as well, now on line 327-330.

Referee #1 (Remarks on code availability):

I did not try to run the code (this is hardly a plug and play exercise). However the explanations about the physical and chemical principles behind the code are very well laid out in the method section and I dont think they require complex numerical schemes.

Referee #3:

The manuscript uses data from the drill hole that sampled Krafla's magma and calculates with numerical models the temperature, pressure, volatile content, and vesicularity evolution of the magma upon drilling.

I am reviewing the revised version and not the original version. This version addresses well the concerns raised by the first round of review. In particular, it clarifies that volatile saturation is not an assumption but a result. It also clarifies that this work is important because it shows in a quantitative way the limits of determining pressure of magma residence based on phase equilibria, melt inclusions, or other thermobarometers.

Detailed comments:

The authors mention the possibility of a magma stored at a less than lithostatic pressure. Could they add a sentence to specify under what circumstances the pressure can be below the lithostatic pressure.

In Elders et al. (2011) they refer to Fournier (1999, <https://doi.org/10.2113/gsecongeo.94.8.1193>) who suggests that plastic rock deformation and failure over a narrow zone results in an abrupt change in pressure from lithostatic to hydrostatic in the magmatic fluids through interaction with the hydrothermal system. We added "due to interaction with the hydrothermal system" to line 77.

l.79-82: sentence starting with "Here, ". I do not understand the sentence, especially where it says "[..] changes in the residual melt/glass between bubbles from which[..]." I suggest to rephrase. Maybe it would be clearer with two sentences.

Rephrased (now line 82-86) to "Here, we expand a bubble growth model for H₂O^[22] to include CO₂, coupled to a model for water species interconversion²³. In addition to the dynamics of bubble growth, a direct output of this model is the residual volatile content in the ~~to simulate the changes in~~

~~the residual~~ melt/glass between bubbles from which we constrain the magma storage conditions of pressure, temperature, and volatile composition against the measured glass chips.”

l. 268: I suggest to be more explicit on how the study produces a tool to engineer magma response and enable safe access via drilling.

Added (now line 235-237) “by forward modeling to find the optimal drilling conditions (e.g. borehole geometry, penetration rate, fluid composition) to inhibit magma ascent up the borehole.”

Extended data figure 2: Please, specify in the caption if the cooling is planar or radial.

Added that this is for the planar geometry.

Catherine Annen

Referee #3 (Remarks on code availability):

Because of time limitation, I did not look at the code in detail, but I read the readme file. It could be clearer by specifying what the different scripts and functions are doing (main.m, Numerical_Model_spec_v2_CO2.m, getFunctions_v2.m, getFunctions_speciation.m). The functions refer to a user manual but I could not see it. I also noticed that readme mentions main.py but the code is main.m.

This code is built on top of an existing architecture by Coumans et al. (2020b), so the reference to a user manual is in the original code comments and would be their user manual. We added to the readme a “Contents” section that specifies each file. Thank you for noticing that the file extension; main is a Matlab, not Python file.

1 **Directly tapping crustal magma reveals storage conditions**

2 Janine Birnbaum^{a,*}, Fabian B. Wadsworth^a, Jackie E. Kendrick^a, Ben Kennedy^b, Paul A.
3 Wallace^a, Marize Muniz da Silva^a, Kai-Uwe Hess^a, and Yan Lavallée^a

4 ^a Ludwig-Maximilians-Universität München

5 ^b University of Canterbury

**The conditions under which magma accumulates and is stored are fundamental to**
**unraveling the processes of crust formation, planetary differentiation, geothermal heat**
**recharge, and volcanic eruptions. Storage pressure and temperature are typically inferred**
**from erupted volcanic products. However, changes during kilometers of magma ascent**
**induce disequilibrium crystallization and vesiculation, and inverting back to storage**
**conditions comes with ~~arguably~~ unresolvable uncertainties. Here, we explore opportunities**
**arising from magma drilling at Krafla volcano (Iceland) to reconstruct real, in situ**
**magmatic conditions for the first time. The findings show that over the ~5 min in which the**
**magma is quenched, vapor bubbles consisting of H₂O and CO₂ exsolve, grow, and resorb,**
**but the changes can be accounted for by ~~careful~~ multiparametric inversion (for chemistry,**
**vesicularity, and vitrification), and that the magma was stored under lithostatic conditions,**
**unlike previous assertions based on classic methods¹. This constraint provides us with the**
**unique pairing of precisely measured depth and pressure on a single magma body, and thus**
**a robust method to improve our understanding of magma storage conditions and evolution.**

Despite their importance, estimates of the pressure, temperature, geometry, and location
of magmatic storage regions vary widely for even the most studied individual volcanic systems.
Geothermobarometers have been applied to volcanic materials to constrain magmatic origins, but
these methods have large uncertainties (~200 MPa or ~10 km) stemming from sparse laboratory
constraints, limits on analytical precision, assumptions of local equilibrium, and inter-dependence
between temperature and pressure². Furthermore, magmatic reservoirs are challenging to image
with geophysical methods due to limitations in resolution and poorly constrained relationships
between lithology and geophysical signals, resulting in typical uncertainties on magma depth of
~0.5-10 km^[3]. As a result, the scientific community lacks consensus on even the fundamentals of
the spatial distribution of melt in the crust⁴.

Deep drilling in hydrothermal fields offers the unique potential for placing tight
constraints on the location, temperature, pressure, and chemistry of melt stored in natural
volcanic systems. Hydrothermal drilling has occasionally intersected magma: dacite in the Puna
geothermal field (Hawaii), trachyte at Menengai (Kenya), and rhyolite at Krafla (Iceland). At
Krafla, the KJ-39 and Iceland Deep Drilling Project-1 (IDDP-1) boreholes directly intersected
magma at ~2500 and 2104 m depth, respectively^{5,6}, which was not anticipated based on coarse
geophysical imaging prior to drilling. Retrospectively, a magma body was recognized during re-
analysis of magnetotelluric (MT) data⁷ and from seismic imaging^{8,9}, which have still been unable
to resolve questions about its lateral and vertical extent.

Silicic glass fragments were recovered from the IDDP-1 borehole which were quenched
 through interaction with drilling fluids^{6,10}. The precisely known recovery depth makes them
 ideally suited to resolve the unknowns of magmatic storage as the melt has not been subject to the
 complex ascent processes that afflict the products of volcanic eruptions. Indeed, the glass
 chemistry has sparked discussion about the origins of the magma from 1) partially molten,
 hydrothermally-altered basaltic crust¹¹ or 2) mantle-derived basalts evolved via fractional
 crystallization¹² and regarding the degree of crustal assimilation^{1,11,13,14}. Various geothermometers
 vary, but generally the mineral assemblage and chemistry can be reconciled with a temperature of
 850-920 °C; however, the corresponding pressure calculated from volatile saturation of the melt
 suggests pressures between ~35 and 45 MPa, significantly below the lithostatic pressure (~50-57
 50 MPa), and above the hydrostatic pressure of the well (~16 MPa)¹.

 **Figure 1: Conflicting petrological constraints on Krafla magma storage conditions.** a)
 Temperature in the borehole⁵, vs geothermobarometric constraints: temperature (red, from augite-
 pigeonite, clinopyroxene-orthopyroxene-palgioclase-magnetite-illmenite, Rhyolite-MELTS¹) and
 pressure (blue, from VolatileCalc^{1,15}). While a trans-crustal mush arrangement of magma storage
 (b) is typically favored⁴, classic views of a basalt-underplated continuous, large rhyolite source
 (c) have also been proposed at Krafla^{12,16} and are not distinguishable based on current
 geophysical observations⁷⁻⁹. d&e) BSE images of IDDP-1 glass chips. Panels b&c adapted
 from¹⁶.

 Although the drilling fluids rapidly cool the magma, it was still subject to decompression
 upon intersection by the drill string¹³, resulting in remobilization; the magma flowed 8 m up the
 well in 9 min^[17]. So we ask, can the IDDP-1 glass be used as a direct record of storage
 conditions? Here, we expand a bubble growth model for H₂O^[18] to include CO₂, coupled to a

model for water species interconversion¹⁹ to simulate the changes in the residual melt/glass
between bubbles from which we constrain the magma storage conditions of pressure,
temperature, and volatile composition.

The glass fragments contain total water of 1.3-2.0 wt%, with an average of 1.8 wt%,
consistent across multiple studies^{1,11,13,15,20,21}, and 50-200 ppm CO₂^[1,15]. Vesicularity in the
quenched glasses is low, with most chips having <6 vol%, although occasionally up to ~15
71 vol%^[13]. Bubble sizes are 1.5-75 μm, with an increase in bubble size with drilling time¹³. Bubble
number densities range between 10^{11.7} and 10¹⁵ 1/m³ which are inferred to have nucleated during
drilling-induced decompression at rates of 10⁵-10⁷ Pa/s^[13,22]. OH/H₂O_m ratios are between
1.68±0.45 and 2.19±0.37, increasing through time¹⁵, and are slightly lower in more vesicular,
2.07±0.20, than in less vesicular, 2.13±0.38, fragments¹.

We explore which P-T paths best reproduce the volatile chemistry and vesicularity of the
IDDP-1 glass. Although the cooling from the drilling fluid should be rapid, we use a thermal
model to seek paths consistent with **measurements of the natural glass**, beginning with one-
dimensional cooling in a planar geometry. We cool the glass via conduction and forced
convection at the surface with supercritical water (400 °C and 16 MPa) where drilling fluid is in
direct contact with the melt. **Thermal analyses** and geospeedometry indicate that the IDDP-1
magma cooled through the glass transition regime (T_g) at 7-80 °C/min with a T_g of ~480 °C^[23].
At a planar interface, **cooling rates far exceed those measured in the glass**. However, at larger
length scales, thermal diffusion is inefficient, such that we can only reproduce the cooling rates in
a narrow region between 0.3 and 1.3 mm from the interface (Supplementary Fig. S1). To produce
a nearly homogeneous glass, **similar cooling rates must be sustained across a large portion of the**
**sampled material. This requires fracturing or fragmentation of the material** to enhanced the
surface area and enable cooling from multiple directions.

If we assume that **magma fragments during the onset or early progression of cooling**, we
should model the thermal evolution instead using a spherical geometry. Under these conditions
we reproduce the measured cooling rates at T_g in the interior of melt fragments of 9-20 mm
radius (Supplementary Fig. S1), a spacing common in **perlitic glass**^{24,25}. This results in a
timescale for cooling from the storage temperature of 900 °C to T_g (at 480 °C) in ~4 min, with a
decelerating cooling rate.

Now we turn to the pressure path experienced by the magma. During drilling the magma
should experience rapid decompression, reflected by high bubble number densities^{13,22}. However,
the **volatile composition and vesicularity of the glass changed only modestly over nine hours of**
**drilling**¹³, suggesting decompression of the magma was also progressive. We suggest **the**
**decompression could be offset at increasing depth by the pressure drop generated by viscous**
**resistance to flow up the borehole**. We estimate the relevant length scale using the Hagen-
Poiseuille equation:

$$102 \quad \Delta P = \frac{8\mu H^2}{R^2} \frac{1}{(1+\phi)^2} \frac{\partial \phi}{\partial t}, \quad (1)$$

using a borehole radius, R, of 0.15 m (12.5" diameter), a melt viscosity, μ, of 3.2×10⁵ Pa s for
magma at 900 °C and 1.8 wt% H₂O, a vesicularity, φ of 1-5%, bubble number density of 10¹¹-

10^{13} 1/m^3 ^[22], and a rate of bubble growth of $5 \text{ }\mu\text{m/s}$ and, resulting in a $\partial\phi/\partial t \sim 1\text{-}10\%/s$. We find a
length scale, H , of $\sim 1.5\text{-}4 \text{ m}$ over which the pressure increases back to the inferred storage
condition (**35-55 MPa**). This has two implications for magma response: 1) the growth of bubbles
in a partially confined space acts to buffer the pressure of magma reservoirs, such that
perturbations from drilling likely only have a local (centimeters to meters) region of influence,
and consequently 2) **most magma may continue to be stored at pressure until the onset of cooling.**

The model outcomes above suggest magma conditions are modulated by cooling and
decompression fronts, whose rates of propagation will be closely linked to, and driven by,
fragmentation. The shear rate for magma flowing in the conduit is $\sim 1\text{-}3 \text{ 1/s}$, insufficient to
produce large-scale shear-driven fragmentation at this viscosity. Similarly, rapid decompression
can result in explosive fragmentation, but requires large pressure drops for low porosity
magmas²⁶. Instead, we suggest **non-explosive fragmentation from thermal shocking** may be the
prevalent fragmentation mechanism, similar to fuel-coolant reactions and the production of
hyaloclastite and other sub-aqueous products²⁷. We use the planar thermal simulations above and
a thermal expansivity of $\sim 6 \times 10^{-6} \text{ 1/}^\circ\text{C}$ to find a region of high strain rate in a propagating front
that moves at $\sim 30 \text{ }\mu\text{m/s}$. Any crack opening would depend on the tensile strength of the material
and be assisted by fluid injection and volume expansion within the cracks. This crack opening
results in an “unzipping” of the magma leading to progressive decompression.

Because we do not simulate the exact fragmentation dynamics, we consider a range of
possible decompression rates from the initial storage pressure (35-55 MPa) to **hydrostatic** (16
125 MPa) over 10 s, 30 s, 1 min, 5 min, 30 min, and 1 hour, corresponding to decompression rates
**5.3-2900 kPa/s**. The **initial conditions are water saturated for a specified molar ratio of water and**
**CO₂ in the co-existing vapor phase**. Fig. 2 shows results with initially 1.8 wt% H₂O_t and 111 ppm
CO₂ at 45 MPa and 900 °C, and OH/H₂O_m of 4.92 (at equilibrium).

Complete decompression from **45 MPa and 900 °C** and **35 MPa and 920 °C** to 16 MPa
produce final vesicularities of 26.2% and 25.6% and water concentrations of 1.34 wt% and 1.32
131 wt%, respectively (Fig. 2), both more vesicular and drier than the measured glass. At magmatic
temperatures, bubble growth is rapid, and only at the highest decompression rates (<5 min) is
there a significant kinetic delay in which bubble growth continues for 20-30 s after the end of
decompression. This suggests that quenching of the glass must begin before the magma is
completely decompressed.

 **Figure 2: Magma vesicularity evolution during different P-T-t regimes requires that cooling**
 **arrests decompression.** Vesicularity along different a) decompression time scales, and cooling
 time scales along b) slow (5 min), and c) rapid (10 s) decompression paths. The shaded regions
 represent the typical ranges of vesicularity measured in the IDDP-1 glass chips, with darker
 shading representing greater abundance amongst measured chips. Line color represents the
 timescales for decompression (blue) and quenching (red).

 We simulate decompression interrupted by cooling at some time offset, Δt_{quench} . When the
 timescales for decompression are short compared to cooling (Fig. 2b) or when cooling occurs
 late, the magma nearly completely degases, but cooling produces resorption of bubbles; while
 decompression reduces solubility, cooling does the reverse and the decompressed magma can
 resorb up to ~ 10 vol% vesicles during cooling. However, to produce final vesicularities within
 the measured range, cooling must be early or rapid compared to decompression (Fig. 2c), in
 which case the final vesicularity is sensitive to the relative timing of cooling onset, and the glass
 locks in a low vesicularity and high water content.

To find the likely storage conditions and decompression paths, we restrict the timing of
 cooling onset to produce the desired final vesicularity of $\sim 5\%$. Because water is the
 volumetrically-dominant species, the final water concentration of the glass and the vesicularity
 are closely related (Fig. 3a,b,e,f,i&j). Together they help constrain the initial water content
 (assuming minimal loss to the surroundings), but there remain a large number of possible
 pressure-temperature paths that are suitable.

 **Figure 3: Simulations of vesicularity need to be combined with water and CO₂ to distinguish**
 **different P-T paths and constrain magma storage to lithostatic pressure.** a,e,&i) Vesicularity,
 b,f&j) total water, and c,g&k) OH/H₂O_m, and d,h&l) CO₂ concentrations along different

[revised manuscript text omitted]

for a linear thermal expansivity, $\alpha = 6 \times 10^{-6} \text{ m}^\circ\text{C}^{[39]}$.

**Water content modeling**

We build upon the work of ref. 18, 19 to numerically model the transport and reaction of
molecular water and hydroxyl dissolved in the glass in response to changing pressure and
temperature conditions. We use the water diffusivity model of ref. 19 in which only molecular
water is able to diffuse, and the diffusivity is a function of both total water and water speciation.
We modify the total water conservation equation of ref. 18 to include diffusion of only molecular
water:

$$296 \quad \frac{\partial H_2O_m}{\partial t} = \frac{1}{r^2} \frac{\partial}{\partial t} \left(r^2 D \frac{\partial H_2O_m}{\partial r} \right), \quad (12)$$

where D is the diffusivity, and H_2O_m is the concentration of molecular water given by:

$$H_2O_m = \frac{W_{H_2O}}{W} X_{H_2O_m}, \quad (13a)$$

$$X_{H_2O_m} + \frac{1}{2} X_{OH} = 1, \quad (13b)$$

 where W_{H_2O} is the molecular weight of water, W is the molecular weight of the hydrous silicate
 melt, $X_{H_2O_m}$ is the molar fraction of molecular water, and X_{OH} is the molar fraction of hydroxyl.
 We solve for conservation of moles of hydroxyl using the reaction equation of ref. 19:

$$\frac{\partial OH}{\partial t} = -2k \left(\frac{OH^2}{K} - H_2O_m X_O \right), \quad (14a)$$

$$X_O = 1 - H_2O_m - OH, \quad (14b)$$

where k is the kinetic reaction rate, K is the equilibrium constant, and X_O is the molar fraction of
 oxygen.

The spherical melt shell is discretized radially with an inner boundary located at the vapor
 bubble wall and an outer boundary determined by the initial bubble number density and
 vesicularity. At the inner boundary in contact with the vapor bubble, we set the molecular water
 concentration according to the equilibrium total water concentration:

$$H_2O_{m,eq} = \begin{cases} H_2O_{t,eq} - OH & OH < H_2O_{t,eq} \\ 1 \times 10^{-15} & OH \geq H_2O_{t,eq} \end{cases}, \quad (15)$$

The outer boundary has a symmetry (no flux) condition. We choose as the initial state pressures
 between 35-55 MPa and temperature between 900-920 °C^[1]. We choose initial water contents of
 1.8 wt% and adjust the starting molar ratio of water and CO₂ in the coexisting vapor phase to
 reach saturation using the mixed solubility model ref. 41. The model is also sensitive to the
 assumed bubble number density which we choose to be 3×10^{14} 1/m³^[13].

**Data Availability**

The simulation results presented in this study will be made available in a public repository
 (Zenodo) upon acceptance.

**Code Availability**

The code developed in this manuscript will be made available in a public repository (Zenodo,
 GitHub) upon acceptance.

**References**

- [1] Zierenberg, R. A., Schiffman, P., Barfod, G. H., Leshner, C. E., Marks, N. E., et al. (2013).
Composition and origin of rhyolite melt intersected by drilling in the Krafla geothermal field,
Iceland. *Contributions to Mineralogy and Petrology*, **165**(2):327–347. doi:10.1007/s00410-012-
0811-z.
- [2] Wieser, P., Gleeson, M., Matthews, S., DeVitre, C., Gazel, E. (Determining the Pressure-
Temperature-Composition (P-T-X) conditions of magma storage (2025). In *Treatise on*
*Geochemistry* (Edition III), **2**:83–151. doi:10.31223/X50M44.
- [3] Lerner, A. H., O’Hara, D., Karlstrom, L., Ebmeier, S. K., Anderson, K. R., Hurwitz S. (2020).
The Prevalence and Significance of Offset Magma Reservoirs at Arc Volcanoes. *Geophysical*
*Research Letters*, **47**(14):1–10. ISSN 19448007. doi:10.1029/2020GL087856.
- [4] Cashman, K. V., Sparks, R. S. J., Blundy, J. D. (2017). Vertically extensive and unstable
magmatic systems: A unified view of igneous processes. *Science*, **355**(6331)
doi:10.1126/science.aag3055.
- [5] Mortensen, A. K., Grönvold, K., Guðmundsson, Á., Steingrímsson, B., Egilson, T. (2010).
Quenched Silicic Glass from Well KJ-39 in Krafla, North-Eastern Iceland. In *Proceedings World*
*Geothermal Congress*, **68**:1-6, Bali, Indonesia.
- [6] Pálsson, B., Hólmgeirsson, S., Guðmundsson, Á, Bóasson, H., Ingason, K., et al. (2014).
Drilling of the well IDDP-1. *Geothermics*, **49**:23–30. doi:10.1016/j.geothermics.2013.08.010.
- [7] Lee, B., Unsworth, M., Árnason, K., Cordell, D. (2020). Imaging the magmatic system
beneath the Krafla geothermal field, Iceland: A new 3-D electrical resistivity model from
inversion of magnetotelluric data. *Geophysical Journal International*, **220**(1):541–567.
doi:10.1093/GJI/GGZ427.
- [8] Kim, D., Brown, L. D., Árnason, K., Gudmundsson, Ó., Ágústsson, K., Flóvenz, Ó. G.
(2020). Magma “bright spots” mapped beneath Krafla, Iceland, using RVSP imaging of reflected
waves from microearthquakes. *Journal of Volcanology and Geothermal Research*, **91**(106365):1–
8. doi:10.1016/j.jvolgeores.2018.04.022.
- [9] Maass, R., Li, K. L., Bean, C. J. (2025). Improving passive reflection seismic imaging in
complex geological settings through site effect reduction: application to Krafla volcano, Iceland.
*Geophysical Journal International*, **241**(2):756–769. doi:10.1093/gji/ggaf072.
- [10] Axelsson, G., Egilson, T., Gylfadóttir, S. S. (2014). Modelling of temperature conditions
near the bottom of well IDDP-1 in Krafla, Northeast Iceland. *Geothermics*, **49**:49–57.
doi:10.1016/j.geothermics.2013.05.003.
- [11] Elders, W. A., Fridleifsson, G., Zierenberg, R. A., Pope, E. C., Mortensen, A. K. et al.
(2011). Origin of a rhyolite that intruded a geothermal well while drilling at the Krafla volcano,
Iceland. *Geology*, **39**(3):231–234, 3 2011. doi:10.1130/G31393.1.
- [12] Rooyackers, S. M., Stix, J., Berlo, K., Petrelli, M., Hampton, R. L. et al. (2021). The

- Origin of Rhyolitic Magmas at Krafla Central Volcano (Iceland). *Journal of Petrology*, **62**(8):1–
27. doi:10.1093/petrology/egab064.
- [13] Saubin, E., Kennedy, B., Tuffen, H., Nichols, A. R., Villeneuve, M. I. et al. (2021). Textural
and geochemical window into the IDDP-1 rhyolitic melt, Krafla, Iceland, and its reaction to
drilling. *Bulletin of the Geological Society of America*, **133**(9-10):1815–1830.
doi:10.1130/B35598.1.
- [14] Seligman, A. N., Bindeman I. N. (2019). The $\delta^{18}\text{O}$ of primary and secondary waters in
hydrous volcanic glass. *Journal of Volcanology and Geothermal Research*, **371**:72–85.
doi:10.1016/j.jvolgeores.2018.12.008.
- [15] Watson, T. (2018). Evolution of magmatic volatiles during drilling into a magma body,
Krafla Iceland. PhD thesis, University of Canterbury, Christchurch, New Zealand.
- [16] Ármannsson, H., Fridriksson, T., Gudfinnsson, G. H., Ólafsson, M., Óskarsson, F.,
Thorbjörnsson, D. (2014). IDDP-The chemistry of the IDDP-01 well fluids in relation to the
geochemistry of the Krafla geothermal system. *Geothermics*, **49**:66–75.
doi:10.1016/j.geothermics.2013.08.005.
- [17] Eichelberger, J. (2020). Distribution and transport of thermal energy within magma–
hydrothermal systems. *Geosciences*, **10**(6):1–28. doi:10.3390/geosciences10060212.
- [18] Coumans, J. P., Llewellyn, E. W., Wadsworth, F. B., Humphreys, M. C., Mathias, et al.
(2020) . An experimentally validated numerical model for bubble growth in magma. *Journal of*
*Volcanology and Geothermal Research*, **402**(107002):1-12.
doi:10.1016/j.jvolgeores.2020.107002.
- [19] Coumans, J. P., Llewellyn, E. W., Humphreys, M. C., Nowak, M., Brooker, R. A. et al.
(2020). An experimentally-validated numerical model of diffusion and speciation of water in
rhyolitic silicate melt. *Geochimica et Cosmochimica Acta*, **276**:219–238.
doi:10.1016/j.gca.2020.02.026.
- [20] Bindeman, I. N., Hudak, M. R., Palandri, J. P., Qi, H., Milovsky, R. et al. (2021). Rhyolitic
and basaltic reference materials for TC/EA analysis: Investigation of water extraction and D/H
ratios. *Chemical Geology*, **583**(120486):1–11. doi:10.1016/j.chemgeo.2021.120486.
- [21] Lowenstern J. B., Pitcher., B. W. (2013). Analysis of H₂O in silicate glass using attenuated
total reflectance (ATR) micro-FTIR spectroscopy. *American Mineralogist*, **98**(10):1660–1668.
doi: 10.2138/am.2013.4466.
- [22] Trewick, L., Tuffen, H., Owen, J., Kennedy, B. M., Eichelberger, J., Zierenberg, R., (2016).
Vesiculation of rhyolite magma in the IDDP-1 borehole at Krafla, Iceland. PhD thesis, Lancaster
University, Lancashire, England.
- [23] Wadsworth, F. B., Vasseur, J., Lavallée, Y., Hess, K. U., Kendrick, J. E et al. (2024). The
rheology of rhyolite magma from the IDDP-1 borehole and Hrafninnuhryggur (Krafla, Iceland)
with implications for geothermal drilling. *Journal of Volcanology and Geothermal Research*,
**455**(108159):1–15. doi:10.1016/j.jvolgeores.2024.108159.

- [24] Davis, B. K., McPhie, J. (1996). Spherulites, quench fractures and relict perlite in a Late
Devonian rhyolite dyke, Queensland, Australia. *Journal of Volcanology and Geothermal*
*Research*, **71**(1):1–11. doi:10.1016/0377-0273(95)00063-1.
- [25] Denton, J. S., Tuffen, H., Gilbert, J. S. (2012). Variations in hydration within perlitised
rhyolitic lavas-evidence from Torfajökull, Iceland. *Journal of Volcanology and Geothermal*
*Research*, **223-224**:64–73. doi:10.1016/j.jvolgeores.2012.02.005.
- [26] Spieler, O., Dingwell, D. B., Alidibirov, M. (2004). Magma fragmentation speed: An
experimental determination. *Journal of Volcanology and Geothermal Research*, **129**(1-3):109–
123. doi:10.1016/S0377-0273(03)00235-X.
- [27] van Otterloo, J., Cas, R. A., Scutter C. R. (2015). The fracture behaviour of volcanic glass
and relevance to quench fragmentation during formation of hyaloclastite and phreatomagmatism.
*Earth-Science Reviews*, **151**:79-116. doi:10.1016/j.earscirev.2015.10.003.
- [28] Zhang, Y., Stolper, E. M., Wasserburg, G. J. (1991). Diffusion of water in rhyolite glasses.
*Geochimica et Cosmochimica Acta*, **55**:441–456. doi:10.1016/0016-7037(91)90003-N.
- [29] Zhang, Y., Stolper, E. M., Ihinger, P. D. (1995). Kinetics of the reaction $H_2O + O = 2OH$ in
rhyolitic and albitic glasses: Preliminary results. *American Mineralogist*, **80**:593–612.
doi:10.2138/am-1995-5-619.
- [30] McIntosh, I. M., Llewellyn, E. W., Humphreys, M. C., Nichols, A. R., Burgisser, A. et al.
(2014). Distribution of dissolved water in magmatic glass records growth and resorption of
bubbles. *Earth and Planetary Science Letters*, **401**:1–11. doi:10.1016/j.epsl.2014.05.037.
- [31] Carey, R. J., Manga, M., Degruyter, W., Gonnermann, H., Swanson, D. et al. (2013).
Convection in a volcanic conduit recorded by bubbles. *Geology*, **41**(4):395–398.
doi:10.1130/G33685.1.
- [32] Watkins, J. M., Manga, M., DePaolo, D. J. (2012). Bubble geobarometry: A record of
pressure changes, degassing, and regassing at Mono Craters, California. *Geology*, **40**(8):699–702.
doi:10.1130/G33027.1.
- [33] Taylor, B. E., Eichelberger, J. C., Westrich, H. R. (1983). Hydrogen isotopic evidence of
rhyolitic magma degassing during shallow intrusion and eruption. *Nature Articles*, **306**:541–545.
- [34] Dobson, P. F., Epstein, S., Stolper, E. M. (1989). Hydrogen isotope fractionation between
coexisting vapor and silicate glasses and melts at low pressure. *Geochimica et Cosmochimica*
*Acta*, **53**:2723–2730. doi:10.1016/0016-7037(89)90143-9.
- [35] Mitchell, S. J., Hudak, M. R., Bindeman, I. N., Carey, R. J., McIntosh, I. M. et al. (2022).
Isotopic signatures of magmatic fluids and seawater within silicic submarine volcanic deposits.
*Geochimica et Cosmochimica Acta*, **326**:214–233. doi: 10.1016/j.gca.2022.03.022.
- [36] Stebbins, J. F., Carmichael, I. S. E., Moret, L. K. (1984). Heat capacities and entropies of
silicate liquids and glasses. *Contributions to Mineralogy and Petrology*, **86**:131–148.
doi:10.1007/BF00381840.

[37] Bouhifd, M. A., Whittington, A., Roux, J., Richet, P. (2006). Effect of water on the heat
capacity of polymerized aluminosilicate glasses and melts. *Geochimica et Cosmochimica Acta*,
**70**(3):711–722. doi:10.1016/j.gca.2005.09.012.

[38] Leshner, C. E., Spera, F. J. (2015). Thermodynamic and Transport Properties of Silicate Melts
and Magma. In *The Encyclopedia of Volcanoes*, 113–141. Elsevier. doi:10.1016/B978-0-12-
385938-9.00005-5.

[39] Bagdassarov, N., Dingwell, D. (1994). Thermal properties of vesicular rhyolite. *Journal of*
*Volcanology and Geothermal Research*, **60**:179–191. doi:10.1016/0377-0273(94)90067-1.

[40] Hess, K.-U. Dingwell, D. B. (1996). Viscosities of hydrous leucogranitic melts: A non-
Arrhenian model. *American Mineralogist*, **81**:1297–1300.

[41] Liu, Y., Zhang, Y., Behrens, H. (2005). Solubility of H₂O in rhyolitic melts at low pressures
and a new empirical model for mixed H₂O-CO₂ solubility in rhyolitic melts. *Journal of*
*Volcanology and Geothermal Research*, **143**:219–235. doi:10.1016/j.jvolgeores.2004.09.019.

**End notes**

**Acknowledgements**

This study was financially supported by the European Research Council (ERC): Magma
Outgassing During Eruptions and Geothermal Exploration (MODERATE No, 101001065).

**Author contributions**

JB: conceptualization, investigation, methodology, software, visualization, writing – original
draft preparation, review and editing; FBW: conceptualization, methodology, writing – review
and editing; JEK: conceptualization, funding acquisition, supervision, writing – review and
editing; BK: writing, review and editing; PAW: writing – review and editing; MMdS: writing –
review and editing; KUH: writing – review and editing; YL: conceptualization, funding
acquisition, methodology, supervision, writing – review and editing.

**Competing interests**

The authors have no competing interests to declare.

**Materials & Correspondence**

Correspondence and requests for materials should be addressed to Janine Birnbaum
(J.Birnbaum@lmu.de)

**Extended Data**

Extended data figure 1: Cooling profile in one direction with A) temperature (°C) and B) cooling
rate (°C/min) colored by time. Grey shaded region shows the approximate glass transition
temperature and stars indicate the time of the glass transition. Cooling rates match natural
samples only at distances of 0.03 and 0.13 cm and timescales between 1-2 min.

Extended data figure 2: Mean A) temperature and B) cooling rate for melt fragments of various
radii Grey shaded region shows the approximate glass transition temperature and stars indicate
the time of the glass transition. To reproduce the observed glass transition temperatures, we
require a fragment size of 9-20 mm radius.

Extended data figure 3: A) Total water, B) OH/H₂O_m, and C) CO₂ dissolved in the melt/glass with
 radial distance away from the bubble center with lighter colors indicating earlier times.
 Decompression and cooling begin at t=0 s, and resorption becomes apparent at t~15s, but by the
 time the melt quenches at 100 s, CO₂ still shows a minimum towards the bubble, total water
 shows a very small maximum towards the bubble, likely within measurement uncertainty for e.g.
 FTIR under most circumstances, and no significant variation in water speciation with distance
 from the bubble.

Files	App Builder	Core Files
		148
> metadata	1.09 KB	149
> environment	199 B	150 t =
> code	57.33 KB	151
getFunctions_speciation.m	9.16 KB	152 300.9417
getFunctions_v2.m	25.99 KB	153
LICENSE	1.04 KB	154
main.m	7.93 KB	155 t =
Numerical_Model_spec_v2_...	12.99 KB	156
run	196 B	157 309.5034
> data Manage	2.72 GB	158
> .codeocean	145 B	159
.gitignore	7 B	160 t =
		323.3226
		[Warning: Imaginary parts of complex X and/or Y arguments ignored.]
		[> In main (line 171)]
		[Warning: Imaginary parts of complex X and/or Y arguments ignored.]
		[> In main (line 179)]
		[Warning: Ignoring extra legend entries.]
		[> In legend>process_inputs (line 590)
		In legend>make_legend (line 319)
		In legend (line 263)
		In main (line 195)]
		[Warning: Ignoring extra legend entries.]
		[> In legend>process_inputs (line 590)
		In legend>make_legend (line 319)
		In legend (line 263)
		In main (line 202)]

```

148
149
150 t =
300.9417
155 t =
309.5034
160 t =
323.3226
[Warning: Imaginary parts of complex X and/or Y arguments ignored.]
[> In main (line 171)]
[Warning: Imaginary parts of complex X and/or Y arguments ignored.]
[> In main (line 179)]
[Warning: Ignoring extra legend entries.]
[> In legend>process_inputs (line 590)
In legend>make_legend (line 319)
In legend (line 263)
In main (line 195)]
[Warning: Ignoring extra legend entries.]
[> In legend>process_inputs (line 590)
In legend>make_legend (line 319)
In legend (line 263)
In main (line 202)]
    
```

```

> Run 9842419/ou...

t =

288.1074

t =

300.9417

t =

309.5034

t =

323.3226

Warning: Imaginary parts of complex X and/or Y arguments ignored.
> In main (line 171)
Warning: Imaginary parts of complex X and/or Y arguments ignored.
> In main (line 179)
Warning: Ignoring extra legend entries.
> In legend>process_inputs (line 590)
In legend>make_legend (line 319)
In legend (line 263)
In main (line 195)
Warning: Ignoring extra legend entries.
> In legend>process_inputs (line 590)
In legend>make_legend (line 319)
In legend (line 263)
In main (line 202)
    
```

Reproducible Run

or launch a cloud workstation

Timeline

Search All

- You have 1 uncommitted change
- Describe what changed:

Edited metadata.yml

Commit Changes
- Julia Hammer ran 00:28:33 Jun 13, 2025 09:20
 - Run 9842419 41.42 MB
 - output 1.38 KB
 - P045_dPdt60_T0650_dTd... 41.41 MB
- about 3 hours ago Capsule duplicated from:
 - Multi-component volatile diffusion into v...

Show Prior History

Reproducibility

Name *

PEER REVIEW COPY: Multi-component volatile diffusion into vapor bubbles

Where do I save my results for Reproducible Run?

Please make sure result files are saved in **/results**, or they will not appear in the computation snapshot in the timeline after a Reproducible Run.

Don't show this message again

Cancel

Done

Markdown is supported.

Tags

Files

Core Files		
> metadata	1.07 KB	✓
> environment	199 B	✓
✓ code	57.33 KB	✓
getFunctions_speciation.m	9.16 KB	✓
getFunctions_v2.m	25.99 KB	✓

```
Metadata run X
#!/usr/bin/env bash
set -ex
# This is the master script for the capsule. When you click "Reproducible Run", the code in
  this file will execute.
matlab -nodisplay -r "addpath(genpath('.')); main"
```

To edit or run this capsule, you need to duplicate it to create your own copy.

Reproducible Run

or launch a cloud workstation

* Required for capsule publication

Metadata

Earth Sciences

PEER REVIEW COPY: Multi-component volatile diffusion into vapor bubbles

Janine Birnbaum, Fabian B. Wadsworth, Jackie E. Kendrick, Ben Kennedy, Paul A. Wallace, Marize Muniz da Silva, Kai-Uwe Hess, Yan Lavallee

This code expands previous work by Coumans et al. (2020) to solve for diffusion of water vapor and CO2 into vapor bubbles, along with reaction between molecular water and hydroxyl groups in silicate melts.

Capsule

c14eff3a-c992-46a3-8bc5-eee8047d423b

Name *

PEER REVIEW COPY: Multi-component volatile diffusion into vapor bubbles

Research Field *

Earth Sciences

Write Preview

Run 9842419/ou... X

```
t =
288.1074

t =
300.9417

t =
309.5034

t =
323.3226
```

```
Warning: Imaginary parts of complex X and/or Y arguments ignored.
> In main (line 171)
Warning: Imaginary parts of complex X and/or Y arguments ignored.
> In main (line 179)
Warning: Ignoring extra legend entries.
> In legend>process_inputs (line 590)
In legend>make_legend (line 319)
In legend (line 263)
In main (line 195)
Warning: Ignoring extra legend entries.
> In legend>process_inputs (line 590)
In legend>make_legend (line 319)
In legend (line 263)
In main (line 202)
```

Reproducible Run

or launch a cloud workstation

Timeline

Search All

You have 1 uncommitted change

Describe what changed:

Edited metadata.yml

Commit Changes

Julia Hammer ran 00:28:33 Jun 13, 2025 09:20

Run 9842419 41.42 MB

output 1.38 KB

P045_dPdt60_T0650_dTd... 41.41 MB

about 3 hours ago Capsule duplicated from :

Multi-component volatile diffusion into v...

Show Prior History

Capsule duplicated. Edit away!

Your copy is saved to your Capsules dashboard.